# Improving the representation of shallow cumulus convection with the simplified-higher-order-closure–mass-flux (SHOC+MF v1.0) approach

**Maria J. Chinita**[1,2], **Mikael Witte**[1,2,3], **Marcin J. Kurowski**[1], **Joao Teixeira**[1,2], **Kay Suselj**[1,6], **Georgios Matheou**[4], and **Peter Bogenschutz**[5]

[1]Joint Institute for Regional Earth System Science and Engineering, University of California Los Angeles, Los Angeles, California, USA

[2]Jet Propulsion Laboratory, California Institute of Technology, Pasadena, California, USA

[3]Department of Meteorology, Naval Postgraduate School, Monterey, California, USA

[4]Department of Mechanical Engineering, University of Connecticut, Storrs, Connecticut, USA

[5]Lawrence Livermore National Laboratory, Livermore, California, USA

[6]Running Tide Technologies, Inc., Portland, Maine, USA

**Correspondence:** Maria J. Chinita (maria.j.chinita.candeias@jpl.nasa.gov)

**Abstract.** CE1 Parameterized boundary layer turbulence and moist convection remain some of the largest sources of uncertainty in general circulation models. High-resolution climate modeling aims to reduce that uncertainty by explicitly attempting to resolve deep moist convective motions. An example of such a model is the Simple Cloud-Resolving E3SM Atmosphere Model (SCREAM) with a target global resolution of 3.25 km, allowing for a more accurate representation of complex mesoscale deep convective dynamics. Yet, small-scale planetary boundary layer turbulence and shallow convection still need to be parameterized, which in SCREAM is accomplished through the turbulent-kinetic-energy-based (TKE-based) simplified higher-order closure (SHOC) – a simplified version of the assumed-double-Gaussian-PDF (probability density function) higher-order-closure method. In this paper, we implement a stochastic-multiplume mass-flux (MF) parameterization of dry and shallow convection in SCREAM to go beyond the limitations of double-Gaussian-PDF closures and couple it to SHOC (SHOC+MF). The new parameterization implemented in a single-column model type version CE2 of SCREAM produces results for two shallow cumulus convection cases (marine and continental shallow convection) that agree well with the reference CE3 large-eddy-simulation data, thus improving the general representation of the thermodynamic quantities and their turbulent fluxes as well as cloud macrophysics in the model. Furthermore, SHOC+MF parameterization shows weak sensitivity to the vertical grid resolution and model time step.

## 1 Introduction

In general circulation models (GCMs), subgrid physical processes need to be parameterized due to the typical horizontal resolutions of GCMs – $\mathcal{O}(10^2)$ km. Traditionally, the turbulent transport in the dry planetary boundary layer (PBL) is represented by a downgradient eddy-diffusivity approach sometimes combined with a countergradient flux term to account for the strong nonlocal transport in the dry convective boundary layer (CBL) (e.g., Deardorff, 1966; Han and Pan, 2011; Teixeira et al., 2004; Holtslag and Moeng, 1991; Stevens, 2000). For shallow cumulus, the transport is often represented by a separate cumulus parameterization based on the mass-flux approach (e.g., Betts, 1973; Tiedtke, 1989; Yoshimura et al., 2015; Beljaars et al., 2018). Such parameterizations often require cloud-base closures and trigger functions. This, combined with the standard GCM modular

structure (i.e., GCMs resort to several independent parameterizations to represent the transport that happens continuously in the real atmosphere) increases uncertainties and biases in GCMs (e.g., Teixeira et al., 2008; Sherwood et al., 2014; Schneider et al., 2017).

During the last 2 decades, unified parameterizations have been proposed and implemented in GCMs to reduce some of the issues associated with conventional modular approaches. Unified parameterizations aim to represent the continuous and evolving turbulent transport across the different PBL regimes, e.g., from dry to shallow cumulus convection, in a consistent manner. Two promising approaches emerged to unify boundary layer turbulence and moist convection: eddy-diffusivity mass-flux (EDMF) methods and higher-order closures (HOCs) based on assumed probability density functions (PDFs). Examples of assumed-PDF schemes include the cloud layers unified by binormals (CLUBB; Golaz et al., 2002) and intermediately prognostic higher-order closure (IPHOC; Cheng and Xu, 2006, 2008), where both schemes assume a double-Gaussian PDF to represent subgrid-scale variability of vertical velocity, temperature, and moisture and, therefore, parameterize PBL turbulence and clouds. A key advantage of HOC-PDF schemes is that cloud macrophysical properties and higher-order moments are diagnosed from the joint PDF in a self-consistent manner. A critical downside is that most HOCs are usually computationally expensive as they require at least seven prognostic equations for second- and third-order moments depending on the chosen PDF. To reduce computational costs, the simplified higher-order closure (SHOC; Bogenschutz and Krueger, 2013) was proposed, for which the higher-order moments needed to construct the PDF are diagnosed instead of prognosed.

The EDMF approach is based on the unification of concepts typically used for the parameterization of boundary layer turbulence (eddy diffusivity) and moist convection (mass flux). It was first proposed for dry convective PBLs (Siebesma and Teixeira, 2000; Teixeira and Siebesma, 2000; Siebesma et al., 2007) and later extended to shallow (e.g., Soares et al., 2004; Neggers, 2009; Rio and Hourdin, 2008; Suselj et al., 2013, 2019a; Tan et al., 2018) and deep convection (Suselj et al., 2019a; Cohen et al., 2020), with the latter representing fully unified parameterizations. In a nutshell, the EDMF approach combines the eddy-diffusivity (ED) and mass-flux (MF) parameterizations, where ED represents the local non-convective mixing and MF represents the nonlocal transport via coherent motions such as updrafts. The stochastic-moist-multiplume mass-flux approach (Suselj et al., 2013, 2019a, b) consists of a fully unified EDMF parameterization of PBL turbulence, as well as dry and moist convection (both shallow and deep) without the usage of trigger functions or cloud-base closures. In the EDMF approach's most recent version, the updrafts are coupled to a simple microphysical scheme allowing for precipitating updrafts. A portion of the updrafts' precipitation falls to the surface, and the remaining forms downdrafts that may lead to cold pools. Although the precipitating EDMF version is somewhat complex, especially in comparison with its non-precipitating version, it is still fairly computational efficient, making it a strong parameterization candidate for any GCM. Several EDMF versions have been successfully implemented and evaluated in both climate GCMs (Kurowski et al., 2019b; Witte et al., 2022) and operational numerical weather prediction models (e.g., Kohler et al., 2011; Suselj et al., 2014, 2021; Han et al., 2016; Olson et al., 2019).

Despite the recent advances in unified parameterizations, parameterized convection remains one of the largest sources of uncertainty in GCMs. Thus, high-resolution climate modeling (e.g., cloud-resolving models, CRMs) is emerging as a pathway to reduce that uncertainty by explicitly resolving some deep convection. An example of such a model is the Simple Cloud-Resolving E3SM Atmosphere Model (SCREAM) with a target global resolution of 3.25 km (Caldwell et al., 2021), allowing for a more accurate representation of complex mesoscale deep convective dynamics. Nevertheless, small-scale PBL turbulence and shallow convection still need to be parameterized, which is accomplished in SCREAM using the HOC-PDF scheme SHOC for computational efficiency.

Recent studies (Firl and Randall, 2015; Fitch, 2019) showed that shallow cumulus convection is not properly represented by HOC-PDF schemes due to limitations of the assumed-double-Gaussian PDF in representing high skewness and kurtosis of the distributions. Moreover, higher-order moments and cloud statistics appear to only be properly represented when a larger number of Gaussians is used in the joint PDF, which increases its already expensive computational cost. An alternative solution was recently proposed in Witte et al. (2022), where CLUBB is combined with multiple stochastic MF plumes, leading to a modified CLUBB+MF parameterization where the plumes represent the extreme tail of the joint distribution, which is not represented by CLUBB (see their Fig. 3). Furthermore, their results showed a large improvement of the higher-order moments for two benchmark shallow cumulus convection cases. Thus, the multiple MF plumes offer a physics-based and cost-effective solution by representing the extreme values of the joint distribution not well captured by the assumed PDF.

Here, our main goal is to improve the representation of shallow cumulus convection in SCREAM by merging SHOC with multiple stochastic MF plumes, thereby creating a unified simplified-higher-order-closure–mass-flux (SHOC+MF) parameterization. In our framework, SHOC represents the local mixing and MF the strong nonlocal mixing. The details of the implementation are described in Sect. 2, and the large-eddy-simulation `CE4` (LES) data are described in Sect. 3. In Sect. 4, we discuss its performance in single-column-model `CE5` (SCM) mode for quasi-steady-state trade-wind maritime shallow cumulus convection and a land diurnal cycle of shallow convection. Lastly, sensitivity tests

to vertical grid resolution and model time steps are also carried out. Conclusions are presented in Sect. 5.

## 2  Methodology

We combine a stochastic-moist-multiplume MF scheme with SHOC (Bogenschutz and Krueger, 2013) in SCREAM. The coupling of MF and SHOC has the potential to improve the representation of the mean thermodynamic structure, higher-order moments (e.g., the turbulent fluxes), and cloud macrophysics quantities by adding the contribution of nonlocal transport during intense convection.

### 2.1  Host model description

SCREAM emerged as one of the next-generation development efforts of the Energy Exascale Earth System Model (E3SM) project led by the U.S. Department of Energy (DOE) to help guide future energy-sector decisions in light of the current long-term trends due to global warming (Golaz et al., 2019; Caldwell et al., 2021). SCREAM is presently still in development but in its final form aims to represent the next generation of global convection-permitting models (GCPMs) by running faster than previous GCPMs due to its performance portability from CPU to GPU machines. To achieve this, the GCM E3SMv1 model serves as a template and is being rewritten from Fortran to C++. SCREAM is based on a nonhydrostatic spectral element dycore and parameterizes turbulence, shallow moist convection, microphysics, radiation, and aerosols (see Caldwell et al., 2021, for a detailed model description). Its target global resolution is 3.25 km.

Here, we use the SCREAM version dyamond2-try1 still written in Fortran and released in October 2020 (https://github.com/E3SM-Project/scream/releases/tag/dyamond2-try1, last access: 20 March 2023), with two modifications: (1) multiplume MF scheme implemented in SHOC's code base and (2) correction of a bug in the SCM spectral element dynamical core that was producing a strong unphysical temperature cold bias. This bug has been fixed in the current development code base (https://github.com/E3SM-Project/E3SM/pull/4027, last access: 20 March 2023).

Our initial assessment of the MF implementation is performed using SCREAM in a SCM framework (Bogenschutz et al., 2020). We are currently migrating the MF component module of the SHOC+MF parameterization to SCREAMv0 (version used in Caldwell et al., 2021), and preliminary results show no significant differences relative to the results presented here.

### 2.2  EDMF parameterization

In weather and climate models, the prognostic equation of the thermodynamic variables depends on the vertical divergence of the turbulent flux in addition to the advective tendencies and diabatic processes:

$$\frac{\partial \overline{\phi}}{\partial t} = -\frac{\partial \overline{w'\phi'}}{\partial z} + F_\phi, \tag{1}$$

where $\overline{\phi}$ represents the prognostic horizontally averaged thermodynamic variable, here taken as the liquid water potential temperature and total water mixing ratio, $\phi = \{\theta_l, q_t\}$; $w$ is the vertical velocity; and the primes denote fluctuations with respect to the mean $\overline{\phi}$. In the convective boundary layer, the turbulent flux corresponds to a combination of small-scale and large-scale coherent turbulent structures and can be decomposed as

$$\overline{w'\phi'} = a_e\overline{w'\phi'}_e + a_e(w_e - \overline{w})(\phi_e - \overline{\phi}) + a_u\overline{w'\phi'}_u + a_u(w_u - \overline{w})(\phi_u - \overline{\phi}), \tag{2}$$

where the subscripts "e" and "u" denote the environment and the strong updrafts, respectively. In the EDMF approach, the following approximations are usually made: (1) the first term is parameterized with the ED approach, (2) the second term is neglected because the environmental and grid-mean values are approximately equal (i.e., $w_e \approx \overline{w}$) following the assumption of small updraft area (i.e., $a_u \ll 1$), and (3) the third term vanishes because the updrafts are assumed horizontally homogeneous and their internal covariances are zero. The fourth term is commonly known as the mass-flux contribution since $M_u \equiv a_u(w_u - \overline{w})$. Thus, Eq. (2) can be simplified to

$$\overline{w'\phi'} = -K_\phi\frac{\partial \overline{\phi}}{\partial z} + M_u(\phi_u - \overline{\phi}), \tag{3}$$

which encapsulates the eddy-diffusivity mass-flux (EDMF) approach (e.g., Siebesma et al., 2007; Suselj et al., 2013). Here, the eddy-diffusivity coefficient, $K_\phi$, is defined according to SHOC's formulation (Bogenschutz and Krueger, 2013), and the MF contribution follows the stochastic-moist-multiplume MF scheme introduced in Suselj et al. (2019a). Thus, the updraft horizontal grid area is partitioned into multiple updrafts, and Eq. (3) is rewritten as

$$\overline{w'\phi'} = -K_\phi\frac{\partial \overline{\phi}}{\partial z} + \sum_{n=1}^N M_n(\phi_n - \overline{\phi}), \tag{4}$$

where $\sum_{n=1}^N M_n(\phi_n - \overline{\phi}) = \sum_{n=1}^N a_n w_n(\phi_n - \overline{\phi})$, $N$ is the user-selected total number of updrafts (here, $N = 40$ updrafts), $a_n$ is the area fraction of the $n$th updraft, and $w_n$ and $\phi_n$ are the vertical velocity and thermodynamic property of the $n$th updraft. The updraft properties are defined according to the updraft model described below (Sect. 2.3).

It is becoming more common to include downdrafts in EDMF parameterizations (e.g., Wu et al., 2020; Han and Bretherton, 2019) mostly due to their relevance to turbulent transport in stratocumulus-topped boundary layers (Chinita et al., 2018; Brient et al., 2019). Despite this, Wu et al. (2020)

showed that the inclusion of updrafts is sufficient to represent the vertical thermodynamic structure and turbulent fluxes of non-precipitating stratocumulus, which is in agreement with the findings reported in Matheou and Teixeira (2019), where the authors showed using LES results that the surface buoyancy and wind shear are as important for turbulence production as cloud-top radiative cooling. Combined with a need for computational efficiency, these recent findings led us to neglect downdrafts in our current MF implementation.

## 2.3 Updraft model

The updraft model closes the multiplume EDMF parameterization and defines the vertical evolution of an updraft's vertical velocity and thermodynamic properties. Here, we follow the updraft model described in Suselj et al. (2019a). Accordingly, at the surface, we release $N$ independent, steady-state buoyancy-driven updrafts with surface vertical velocities sampled from the right tail of an assumed-Gaussian PDF, with values ranging between $w_{\min}$ and $w_{\max}$, here defined as $1.5\sigma_w < w_n < 3\sigma_w$, where $\sigma_w$ is the vertical velocity standard deviation (note that the interval $[1.5\sigma_w, 3\sigma_w]$ corresponds to a total updraft surface fraction area equal to 6.65 %, in agreement with the sensitivity analysis to the surface updraft area presented in Suselj et al., 2019a). The tail of the velocity PDF, i.e., the interval $[1.5\sigma_w, 3\sigma_w]$, is discretized into $N$ equidistant bins, and the mean vertical velocity value of each bin is associated with a corresponding updraft ($N$ is the total number of updrafts). The surface thermodynamic properties of each updraft are computed by integrating the joint-normal PDF ($\theta_{lu}$, $q_{tu}$ TS1, $w_u$) over the updraft's velocity bin (see Suselj et al., 2019b, for details on the joint-normal PDF characterization). Here, we use $N = 40$ updrafts. The number of updrafts was chosen based on a sensitivity analysis of SHOC+MF to its value in which SHOC+MF showed weak sensitivity to $N > 30$ updrafts (not shown). Note that a small number of updrafts can produce noisier results due to the lateral entrainment's stochasticity (Suselj et al., 2019b).

The vertical evolution of the $n$th updraft depends on surface properties and lateral entrainment as follows:

$$\frac{\partial \phi_n}{\partial z} = \varepsilon_n \left(\overline{\phi} - \phi_n\right), \tag{5}$$

where $\phi = (\theta_l, q_t)$ and $\varepsilon_n$ is the entrainment rate of the $n$th updraft. Thus, Eq. (5) represents the dilution of $\phi_n$ by lateral entrainment of environmental air $\overline{\phi}$; the environmental air properties are assumed equal to the grid-mean values following Kurowski et al. (2019a). The vertical velocity of the $n$th updraft is determined by TS2

$$\frac{\partial w_n^2}{\partial z} = a_w B_n - b_w \varepsilon_n w_n^2, \tag{6}$$

where $a_w = 1$ and $b_w = 1.5$ are constants and $B_n$ is the updraft's buoyancy given by $B_n = g\left(\theta_{v,n}/\overline{\theta}_v - 1\right)$, where $\theta_v$

is the virtual potential temperature. The boundary condition values needed to integrate Eqs. (5) and (6), i.e., the surface thermodynamic properties ($w_n|_s$, $\theta_{v,n}|_s$, $q_{t,n}|_s$), are computed as in Suselj et al. (2019a), and their standard deviation values ($\sigma_w$, $\sigma_{\theta_v}$, $\sigma_{q_t}$) follow Suselj et al. (2019b). Note that $\theta_{l,n}|_s$ is defined with respect to $\theta_{v,n}|_s$ as $\theta_{l,n}|_s = \theta_{v,n}|_s/(1 + 0.61 q_{t,n}|_s)$ assuming $\theta_{l,n}|_s \equiv \theta_n|_s$ (subscript "s" denotes surface). The numerical discretization of Eqs. (5) and (6) follows that described in Suselj et al. (2014).

Lastly, the lateral entrainment of the $n$th updraft is defined as a stochastic process (Romps and Kuang, 2010; Suselj et al., 2019b): TS3

$$\varepsilon_n = \frac{\varepsilon_0}{\Delta z} \mathcal{P}_n\left(\frac{\Delta z}{L_\varepsilon}\right), \tag{7}$$

where $\varepsilon_0$ is the fraction of entrained mass flux during each entrainment event, here set to $\varepsilon_0 = 0.2$; $\mathcal{P}_n$ is a random number drawn from the Poisson distribution that represents the number of entrainment events for a given average event frequency equal to $L_\varepsilon$, and $\Delta z$ is the thickness of the respective layer. Note that we evaluate $\mathcal{P}_n$ and $\varepsilon_n$ for each updraft independently. Following Suselj et al. (2019b), the entrainment length scale is defined as a function of the depth of the CBL including the cloud layer when present, $h_{CBL}$:

$$L_\varepsilon = a\sqrt{h_{CBL}}, \tag{8}$$

where $a = 1.25$ m$^{1/2}$ is a constant and $h_{CBL}$ is defined as the model level where the vertical heat flux vanishes ($\overline{w'\theta_l'} \approx 0$). In agreement with previous studies (e.g., Böing et al., 2012; Takahashi et al., 2021), the entrainment length scale $L_\varepsilon$ is larger for deeper clouds (i.e., higher $h_{CBL}$) as these tend to be wider and thus better protected from the environment, leading to smaller entrainment rates. Note that diagnosing $L_\varepsilon$ as the square root of $h_{CBL}$ allows for continuous adjustment of $\varepsilon_n$ as a function of the CBL state; i.e., the entrainment rate is reduced for deeper CBLs, allowing the updrafts to reach higher vertical levels and vice versa for shallower CBLs, which is particularly important to represent the strong diurnal cycle over land while remaining insensitive to small oscillations of $h_{CBL}$.

Condensation in each updraft takes place if the updraft water vapor reaches saturation. The MF contribution to the total cloud fraction corresponds to the sum of the area fraction of the updrafts that condense, and the MF contribution to the total cloud water is defined as the area average of the cloud water of all moist updrafts.

## 2.4 SHOC

In SCREAM, boundary layer turbulence and moist shallow convection are parameterized by SHOC (Bogenschutz and Krueger, 2013). SHOC is considered a simplified assumed-PDF-based scheme because the second-order moments needed to construct the PDF are diagnosed instead of

prognosed to increase computational efficiency. Accordingly, the turbulent fluxes $\overline{w'\theta_l'}$ and $\overline{w'q_t'}$ are estimated following an eddy-diffusivity approach:

$$\overline{w'\phi'} = -K_\phi \frac{\partial \overline{\phi}}{\partial z}, \tag{9}$$

where $\phi = \{\theta_l, q_t\}$ and $K_\phi$ represents the eddy-diffusivity coefficient for heat. It is important to note that SHOC has been updated since Bogenschutz and Krueger (2013) to improve numerical stability and overall performance to better represent the various regimes present in a GCM. For instance, the formulation of the turbulence length scale has been revised and now follows a continuous formulation instead of two separate definitions for the sub-cloud and cloud layers as documented in Bogenschutz and Krueger (2013). Nevertheless, the SHOC version used in SCREAM exhibits similar scientific performance to the original formulation. For completeness, the turbulence length scale is defined as

$$L = \frac{1}{l_c}\sqrt{8\left[\frac{1}{\tau\sqrt{e}kz} + \frac{1}{\tau\sqrt{e}L_\infty} + 0.01\delta\frac{N^2}{e}\right]^{-1}}, \tag{10}$$

where $k$ is the von Karman constant, $e$ is the turbulent kinetic energy, and $l_c = 0.5$ is a tunable length scale factor. The constant $\delta$ is defined as $\delta = 1$ if the Brunt–Väisälä frequency $N^2 > 0$ or $\delta = 0$ if $N^2 \leq 0$, where $N^2 = \frac{g}{\overline{\theta_v}}\frac{\partial\overline{\theta_v}}{\partial z}$. The asymptotic value of the length scale $L_\infty$ is defined following Blackadar (1962) as $L_\infty = 0.1\int_0^\infty \sqrt{e}z\mathrm{d}z/\int_0^\infty \sqrt{e}\mathrm{d}z$. Lastly, the eddy turnover timescale $\tau$ is defined as $\tau = h/w_*$, where $h$ is the boundary layer depth calculated according to Holtslag and Boville (1993), and the convective velocity scale $w_*^3 = 2.5\frac{g}{\overline{\theta_v}}\text{TS4}\int_0^h \overline{w'\theta_v'}\mathrm{d}z$. If the boundary layer is stable (i.e., $w_*^3 < 0$), then $\tau = 100$ s.

## 2.5 Coupling of SHOC and multiplume mass-flux parameterizations

We implement the stochastic-multiplume MF scheme in SCREAM by coupling it to SHOC. Thus, the multiplume MF contribution (second term of the right-hand side of Eq. 11) is added to SHOC's numerical solver for the mean thermodynamic variables, $\phi = (\theta_l, q_t)$, according to the following one-dimensional prognostic equation: TS5

$$\begin{aligned}\frac{\partial\phi}{\partial t} &= -\frac{\partial\overline{w'\phi'}}{\partial z}\\&= -\frac{\partial}{\partial z}\left(\underbrace{-K_\phi\frac{\partial\overline{\phi}}{\partial z}}_{\text{SHOC}} + \underbrace{\sum_{n=1}^N a_n(w_n - \overline{w})(\phi_n - \overline{\phi})}_{\text{MF}}\right),\end{aligned} \tag{11}$$

where $K_\phi$ is the eddy-diffusivity coefficient, $a_n$ is the area fraction of the $n$th updraft, $w_n$ and $\phi_n$ are the vertical velocity and the $\phi$ value in the $n$th plume, and the overbar denotes

a grid-mean value. The SHOC term (first term of the right-hand side of Eq. 11) represents the time tendency of $\phi$ due to the downgradient diffusion of the mean field, and the MF term takes into account the nonlocal transport due to strong convection as discussed in Sect. 2.2. The prognostic Eq. (11) is discretized according to the semi-implicit forward-in-time centered-in-space scheme and solved using the Richtmyer and Morton (1967) method (see Kurowski et al., 2019b, for the discretized form of Eq. 11). Note that the surface boundary conditions of Eq. (11) (i.e., the surface fluxes of the thermodynamic variables, $\overline{w'\phi'}_s$) are either calculated by the surface layer parameterization or are prescribed and not modified by the MF component. Results of the coupling of MF and SHOC are denoted as "SHOC+MF" in the next sections.

The shortwave and longwave radiation and the large-scale cloud microphysics parameterizations are switched off for these experiments – basically because these processes are believed to be of secondary importance for these shallow convection case studies and as such are also off for the LES experiments. However, cloud fraction and water are calculated at every model level for diagnostic purposes. In SCREAM, the cloud macrophysical properties cloud fraction and liquid water mixing ratio are estimated using the SHOC PDF. Here, the combination of the grid-mean cloud properties is done as a simple weighted sum of the SHOC and MF contributions:

$$\text{CF} = \text{CF}_{\text{SHOC}} + \sum_{n=1}^N a_n(q_{l_n} > 0), \tag{12}$$

$$\overline{q_l} = a_e q_{l_{\text{SHOC}}} + \sum_{n=1}^N a_n q_{l_n}, \tag{13}$$

where $\text{CF}_{\text{SHOC}}$ and $q_{l_{\text{SHOC}}}$ are diagnosed from SHOC's assumed-double-Gaussian distribution, and $a_n$ and $q_{l_n}$ are the fractional area and condensate loading of the $n$th updraft. Note that, in practice, although the algorithm also imposes $\text{CF} \leq 1$, this value is not reached in these simulations because of the low values typical of these shallow convection case studies. These low cloud values, the overall cloud vertical structure of these shallow convection regimes, and the fact that the radiation parameterization is off are the key reasons for not using more complex cloud overlap algorithms in these estimates.

## 2.6 Single-column-model simulations

CE6 The ability of the unified SHOC+MF parameterization to represent shallow cumulus boundary layers has been investigated by simulating benchmark cases including the shallow cumulus Barbados Oceanographic and Meteorological Experiment (BOMEX; Siebesma et al., 2003), quasi-steady-state warm maritime shallow convection over the Atlantic Ocean in June 1969, and the Atmospheric Radiation Measurement (ARM) shallow cumulus case (Brown et al., 2002), diurnal cycle of warm shallow convection over land at the Southern Great Plains site of the ARM program on 21 June 1997. The two cases were simulated using SCREAM in a SCM framework in which we used the TS6 intensive-

observation-period CE7 (IOP) forcing files available in the E3SM SCM library (Bogenschutz et al., 2020) with prescribed horizontal large-scale forcing and surface turbulent fluxes. It is worth noting that we modified the ARM case forcing file to run the model with a 30 min time step (i.e., the ARM forcing file available in the E3SM SCM library contains values at every 20 min). Also, the SCM reads the wind information from the forcing file at every host model time step; however, for the ARM case, the large-scale advective tendencies of the winds were not available when the case was set up (Brown et al., 2002), and, consequently, the time-varying $u$ profiles in the forcing file were set equal to the initial profiles which are constant with height. Resetting the $u$ profile to the initial vertically constant profile at every host model time step interferes with the development of the turbulent-kinetic-energy (TKE) field through the shear production term. To circumvent this issue, we replaced the $u$ profiles in the forcing file with the $u$ profiles from our LES reference data; the meridional wind component $v$ is zero in the ARM case. Note that this is specific to the SCM used here and to the ARM case as the large-scale advective tendencies of the winds were not available when the ARM case was set up (Brown et al., 2002).

We kept the default host model setup but deactivated the deep-convection, large-scale microphysics and radiation schemes to allow for a more straightforward comparison with our LES reference data. The dynamic and physics time steps are equal to 30 and 5 min, respectively. For our initial implementation and performance evaluation, we used a 72-layer vertical grid (L72) with 21 levels resolving the bottom 3 km. In Sect. 4.3, we conduct tests to quantify sensitivity to the vertical grid resolution and to the time step using BOMEX. Thus, for the vertical grid resolution, we assess the sensitivity of the results using L72 and a relatively finer 128-layer vertical grid (L128) with twice as many grid cells resolving the bottom 3 km (40 grid cells). For the time step sensitivity, we compare the results using the L128 and dynamics and physics time steps equal to 30 and 5 min (300 s), respectively, with dynamics and physics time steps both equal to 75 s, which resembles the configuration used in Caldwell et al. (2021) for the first global results of SCREAM in convection-permitting mode ($\Delta x = 3.25$ km).

## 3 Large-eddy-simulation model

CE8 We evaluate our SHOC+MF parameterization by comparing it to LES output of the same benchmark cases. These LES reference data are acquired with the LES model of Matheou and Chung (2014). Table 1 summarizes the LES runs and their configurations. The computational domain is doubly periodic in the horizontal directions, and all grids are uniform and isotropic ($\Delta x = \Delta y = \Delta z$). The simulations have different domain sizes in the vertical adjusted to their respective boundary layer depths. A Rayleigh damping layer is imposed near the domain top to mitigate gravity wave reflection, and the surface turbulent fluxes are prescribed as in the SCREAM SCM. The momentum and scalar advection terms are discretized according to the sixth-order fully conservative centered scheme of Morinishi et al. (1998) adapted for the anelastic approximation (Matheou et al., 2016). The subgridscale (SGS) turbulence is represented by the buoyancy-adjusted stretched-vortex SGS model (Chung and Matheou, 2014). Precipitation is neglected in the LES model according to the case descriptions (Brown et al., 2002; Siebesma et al., 2003), and all water condensate is assumed suspended using an all-or-nothing saturation adjustment scheme based on the local grid-mean state. The simulations are carried out in the frame of reference of the domain-mean horizontal wind to reduce numerical errors (Lamaakel and Matheou, 2021). The LES model has been successfully used in previous studies spanning several meteorological conditions (Chung et al., 2012; Matheou and Chung, 2014; Matheou, 2018; Matheou and Teixeira, 2019; Couvreux et al., 2020; Chinita et al., 2022a, b).

## 4 Results

We compare the results of SHOC and SHOC+MF against the LES reference data for the benchmark cases listed in Sect. 2.6. Note that in SHOC+MF, we reduced SHOC's turbulence mixing length scale relative to SHOC alone to prevent excessive mixing. This was done by increasing the tunable length scale factor $l_c$ from 0.5 to 1 in Eq. (10). Thus, $l_c = 0.5$ in the SHOC experiments, and $l_c = 1$ in the SHOC+MF experiments. All other model and parameterization configurations were kept the same for all simulations shown here. Lastly, for the simulations of SHOC alone, we used the same tunable constants from the global high-resolution simulation presented in Caldwell et al. (2021).

### 4.1 Trade-wind maritime shallow cumulus

Figure 1 shows results for BOMEX averaged over simulated hours 4 to 6. The thermodynamics profiles are generally similar but with some noticeable differences: SHOC is colder above cloud base and warmer near the cloud top relative to the LES (Fig. 1a), and it is moister above cloud base and drier near the cloud top (Fig. 1b). This is because SHOC mixes excessively up to $\sim 1$ km and does not reproduce a shallow cumulus layer (Fig. 1d and e). Consequently, moisture does not reach the levels where the cloud top should be located and instead it gets trapped between 0.5 and 1 km. In contrast, the turbulent transport of SHOC+MF is very similar to the LES, leading to comparable thermodynamic profiles, except near the cloud top ($\sim 1.5$ km), where SHOC+MF mixes slightly less, leading to a drier (warmer) layer relative to the LES. The partitioning of turbulent transport between local and nonlocal mixing in SHOC+MF is similar to previous EDMF studies

**Table 1.** Summary of the cases simulated. The details of each case setup are described in the references (second column). Here, $L_{x,y}$ and $L_z$ are the horizontal and vertical domain lengths, $N_{x,y}$ and $N_z$ are the number of horizontal and vertical grid points, and $\Delta x$ is the grid spacing.

| Case | Reference | $L_{x,y}$ (km) | $L_z$ (km) | $N_{x,y}$ | $N_z$ | $\Delta x$ |
|------|-----------|------|------|------|------|------|
| Maritime shallow convection | Siebesma et al. (2003) | 20.48 | 3.0 | 1024 | 150 | 20 |
| Continental shallow convection | Brown et al. (2002) | 20.48 | 4.4 | 1024 | 220 | 20 |

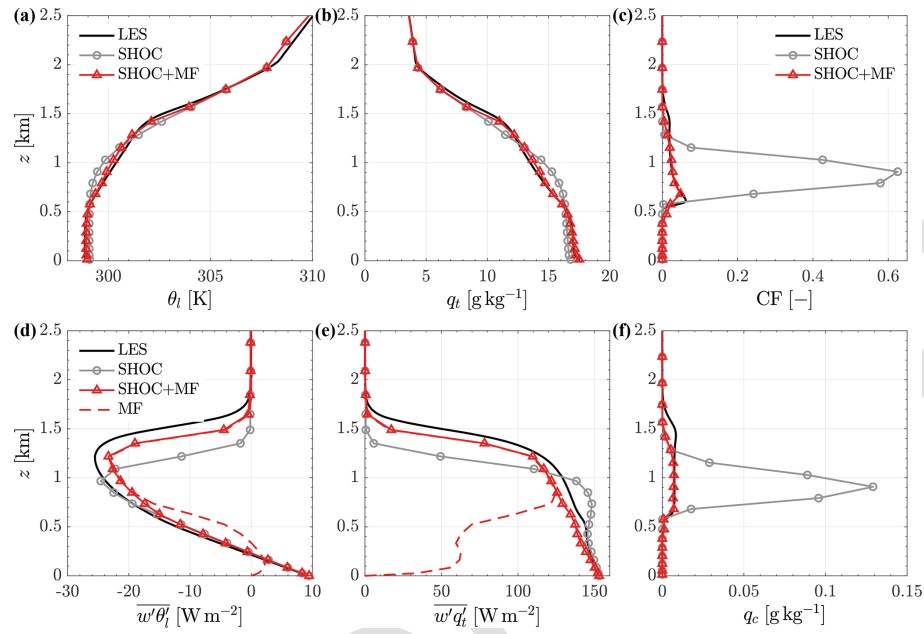

**Figure 1.** Vertical profiles of **(a)** liquid water potential temperature, **(b)** total water mixing ratio, **(c)** cloud fraction, **(d)** turbulent heat flux, **(e)** turbulent moisture flux, and **(f)** cloud water mixing ratio for LES (solid black line), SHOC (solid grey line), SHOC+MF (solid red line), and MF (dashed red line) for the BOMEX case. The profiles correspond to a time average over $t = 4$–$6$ h.

(e.g., Suselj et al., 2013; Kurowski et al., 2019b); i.e., the local mixing dominates in the subcloud layer, and MF takes over in the cloud layer.

Because of the excessive humidity between 0.5 and 1 km in SHOC, the cloud fraction and liquid water mixing ratio are overestimated relative to the LES (Fig. 1c and f). On the other hand, SHOC+MF captures the profiles of both cloud fraction and cloud liquid water content fairly well due to the adequate vertical distribution of the thermodynamic quantities. Note that the cloud fraction and liquid water content of SHOC+MF shown in Fig. 1c and f are calculated as the sum of SHOC and MF contributions.

A key aspect in simulating shallow cumulus with an MF-type parameterization like SHOC+MF is the accurate representation of the moist updraft properties (i.e., updraft area, vertical velocity, and the excess of moist conserved variables). Figure 2 shows the moist updraft properties of the mass flux of SHOC+MF and the respective LES values based on cloud ($q_l > 1 \times 10^{-5}$ kg kg$^{-1}$) and cloud core ($q_l > 1 \times 10^{-5}$ kg kg$^{-1}$, $w > 0$, $\theta_\upsilon > \langle\theta_\upsilon\rangle$, where the angle brackets denote the instantaneous horizontal average of the LES domain) samplings (Siebesma and Cuijpers, 1995). Since the

SHOC+MF turbulent transport is controlled mostly by MF in the cloud layer (dashed profiles in Fig. 1d and e), the moist updraft properties should lie close to the LES cloud and cloud core values (Couvreux et al., 2010; Suselj et al., 2013; Kurowski et al., 2019b). The SHOC+MF updraft area agrees reasonably well with the LES values, especially when considering the relatively coarse vertical grid used here, resulting in just two grid levels to resolve the sharp increase near the cloud base. The vertical velocity and the excess of updraft moist conserved variables relative to the grid-mean values of SHOC+MF are close to the LES profiles, except in the middle of the cloud layer ($\sim 1$ km), where the $\theta_{lu}$ and $q_{tu}$ excesses are underestimated. This is due to the slight overestimation of the SHOC+MF grid-mean $q_t$ (Fig. 1b) relative to the LES. Nevertheless, the SHOC+MF moist updraft properties agree well with the LES, which confirms the suitable behavior of our MF scheme. Note that the updraft (second Gaussian) moist properties of the SHOC's PDF are not shown because they are quite small (e.g., maximum $w_u \approx 0.3$ m s$^{-1}$) and vanish around 800 m in agreement with Fig. 1d–e, where the MF contribution makes up the total turbulent fluxes.

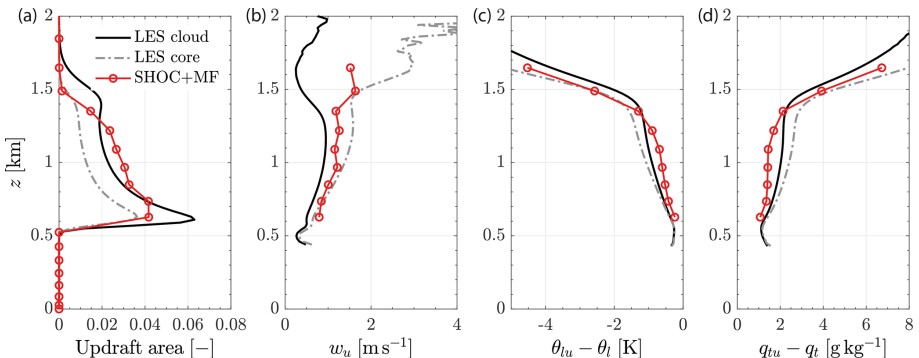

**Figure 2.** Vertical profiles of moist updraft properties for the BOMEX case. **(a)** Updraft area, **(b)** updraft vertical velocity, and excess relative to the grid-mean values of **(c)** liquid water potential temperature ($\theta_{lu} - \overline{\theta}_l$) and **(d)** total water mixing ratio ($q_{tu} - \overline{q}_t$). The solid black lines correspond to the LES cloud sampling, the dashed grey lines to the LES cloud core sampling, and the solid red lines to SHOC+MF. The profiles correspond to a time average over $t = 4$–$6\,$h.

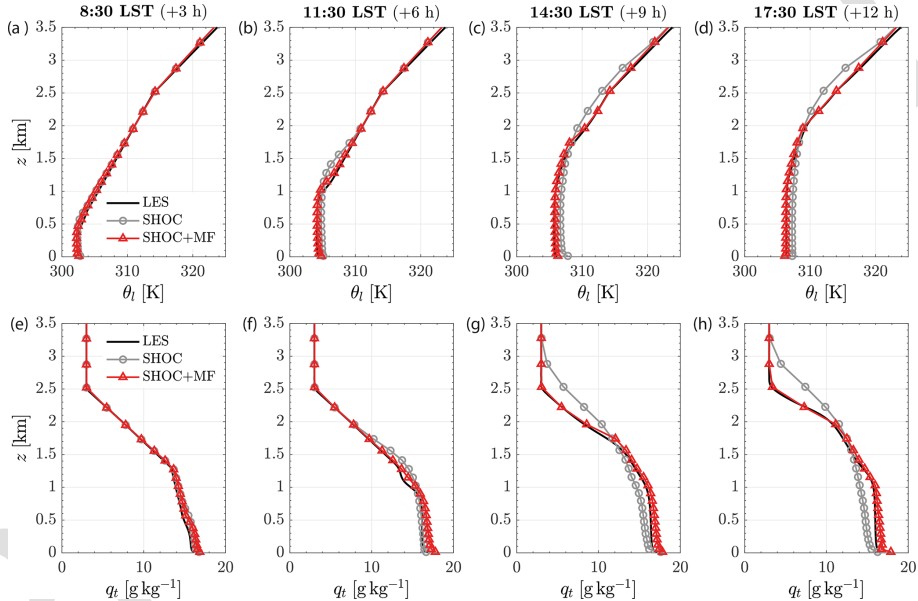

**Figure 3.** Vertical profiles of **(a–d)** liquid water potential temperature and **(e–h)** total water mixing ratio for LES (solid black line), SHOC (solid grey line), and SHOC+MF (solid red line) for the ARM shallow cumulus case. The label "HH:MM LST ($+X$ h)" shows the local standard times (LST) and the hour mark relative to the start of the simulation. The profiles correspond to hourly averages around the hour marked on each column (HH:MM LST $\pm30\,$min).

## 4.2 Continental shallow cumulus

The ARM shallow cumulus case represents a diurnal cycle of warm convection over land at the Southern Great Plains site of the ARM program on 21 June 1997 (Brown et al., 2002). The case starts with a morning transition at 11:30 UTC (5:30 LST – local standard time) from a stable boundary layer (negative surface heat flux until 1.5 simulated hours) to a fully developed CBL with a top close to 2.5 km at around 21:00 UTC (15:00 LST). The case represents a typical buoyancy-driven shallow cumulus case where convection is primarily forced by the surface sensible and latent heat fluxes. The nonstationary conditions of the ARM

shallow cumulus case make it more challenging to properly simulate than the quasi-steady-state BOMEX case.

Figure 3 shows that the diurnal evolution of hourly mean thermodynamic profiles is well represented by SHOC+MF, whereas SHOC produces a warm (dry) bias in the subcloud layer and a cold (moist) bias near the cloud top. To illustrate the magnitude and temporal evolution of these biases, panels a and b of Fig. 4 show the temperature and moisture differences relative to the LES fields, and by the end of the simulation, the temperature (moisture) bias exceeds $0.5\,$K ($1\,$g kg$^{-1}$) in the subcloud layer and $1.5\,$K ($4\,$g kg$^{-1}$) near the cloud top. On the other hand, SHOC+MF is able to reproduce the diurnal evolution of the PBL and cloud

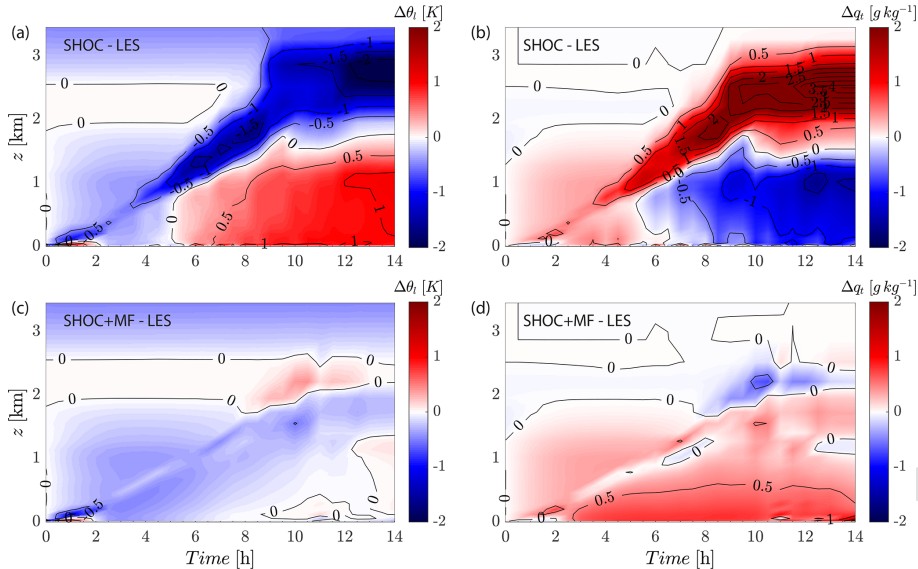

**Figure 4.** Time–height plot of liquid water potential temperature differences $\Delta\theta_l$ between SCM and LES for **(a)** SHOC and **(c)** SHOC+MF as well as total water mixing ratio differences $\Delta q_t$ for **(b)** SHOC and **(d)** SHOC+MF for the ARM shallow cumulus case. The LES temporal and vertical grids were interpolated to the SCREAM grids before calculating the differences.

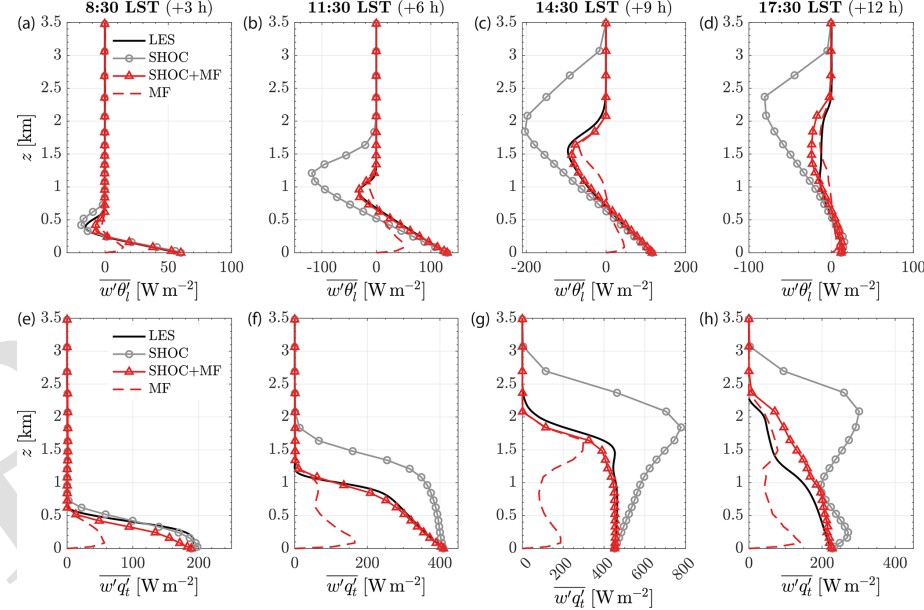

**Figure 5.** Vertical profiles of turbulent fluxes of **(a–d)** liquid water potential temperature and **(e–h)** total water mixing ratio for LES (solid black line), SHOC (solid grey line), SHOC+MF (solid red line), and MF (dashed red line) for the ARM shallow cumulus case. The label "HH:MM LST (+$X$ h)" shows the local standard times (LST) and the hour mark relative to the start of the simulation. The profiles correspond to hourly averages around the hour marked on each column (HH:MM LST ±30 min).

layer remarkably well for both thermodynamic quantities. The largest bias is located near the surface for the total water mixing ratio with maximum deviations from LES around +0.5 g kg$^{-1}$ (Fig. 4d).

Figure 5 shows the diurnal evolution of hourly mean profiles of the turbulent fluxes of liquid water potential temper-

ature (top row) and total water mixing ratio (bottom row). In agreement with the vertical distribution of the thermodynamic quantities, SHOC+MF represents both turbulent transports fairly well, whereas SHOC produces excessive turbulent mixing and consequently places the cloud top near 3 km at 20:30 UTC – about 1 km deeper than the LES. Fig-

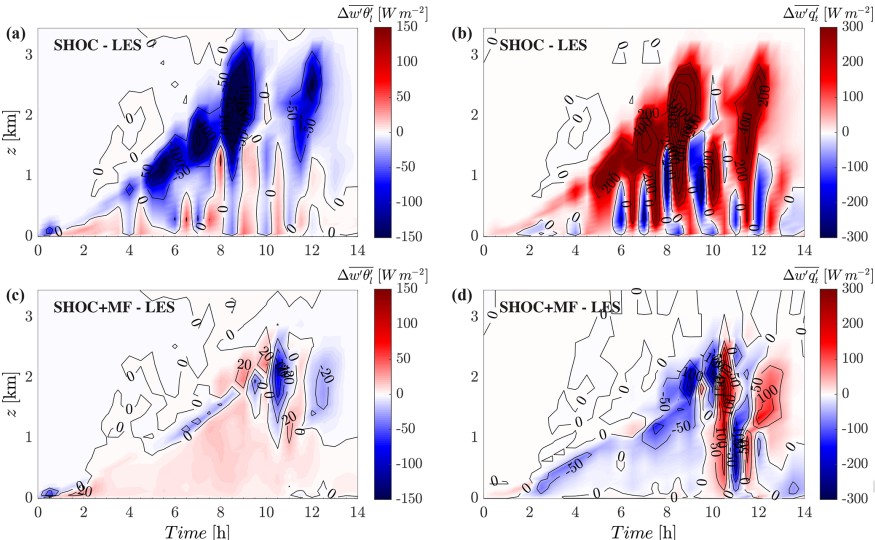

**Figure 6.** Time–height plot of liquid water potential temperature turbulent flux differences $\Delta\overline{w'\theta_l'}$ between SCM and LES for **(a)** SHOC and **(c)** SHOC+MF as well as total water mixing ratio turbulent flux differences $\Delta\overline{w'q_t'}$ for **(b)** SHOC and **(d)** SHOC+MF for the ARM shallow cumulus case. The LES temporal and vertical grids were interpolated to the SCREAM grids before calculating the differences.

ure 6 shows the differences between the temperature and moisture turbulent fluxes of SHOC and SHOC+MF relative to the LES. Both temperature and moisture panels (a–b) confirm SHOC's excessive mixing and cloud layer deepening and also reveal oscillations, particularly on the $\overline{w'q_t'}$ field after hour 5. These oscillations may be due to the eddy turnover timescale $\tau$ used in the calculation of SHOC's turbulence mixing length scale (Eq. 10) in this SCREAM version since these are not present when a constant timescale is used (e.g., $\tau = 400$ s; not shown). However, SHOC+MF is able to reproduce the turbulent fluxes without these oscillations (Fig. 6c–d) while using the dynamic timescale, matching the LES reasonably well, except for the last 4 h of simulation where $\overline{w'q_t'}$ decreases slower (faster) than the LES in the upper half (lower half) of the boundary layer (Fig. 5h).

Note that the turbulent transport partition between local and nonlocal mixing in SHOC+MF is similar to BOMEX when the cloud layer forms (from simulated hour 5 to hour 12); i.e., the transport is mostly controlled by the local mixing in the subcloud layer, whereas MF dominates in the cloud layer (e.g., dashed red profiles in Fig. 5c and g). Before cloud formation, the local mixing contribution to the turbulent transport is larger and the MF contribution is only significant near the surface. A similar behavior was observed for a dry convection case (case 1 of Siebesma et al., 2007 – not shown), where SHOC properly represented the turbulent transport including the PBL growth but developed a warm bias near the surface – this is a typical pattern of ED-type schemes without MF (e.g., Teixeira and Cheinet, 2004; Siebesma et al., 2007; Witek et al., 2011). The inclusion of MF partially reduced this warm bias by slightly enhancing

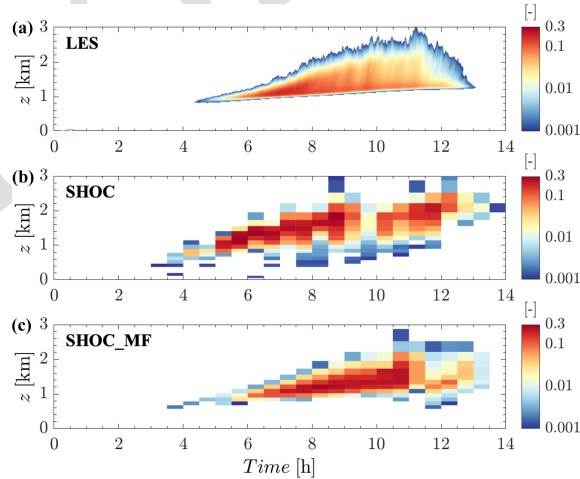

**Figure 7.** Time–height plot of **(a)** LES, **(b)** SHOC, and **(c)** SHOC+MF cloud fraction for the ARM shallow cumulus case. The LES cloud fraction corresponds to the cloud sampling definition (i.e., $q_l > 1 \times 10^{-5}\,\mathrm{kg\,kg^{-1}}$). Cloud fraction values $< 0.001$ are masked and not plotted here.

the turbulent mixing near the surface whilst adjusting its contribution to negligible values away from the surface.

Figure 7 shows the temporal evolution of the cloud fraction. The LES cloud fraction field is smoother than SHOC and SHOC+MF due to the LES higher temporal resolution ($\Delta t = 1$ min vs. 30 min). Nevertheless, SHOC+MF cloud cover (Fig. 7c) roughly follows the LES, except between hours 10 and 12 because of its excessive turbulent flux in the cloud layer (Fig. 5h), resulting in significantly higher

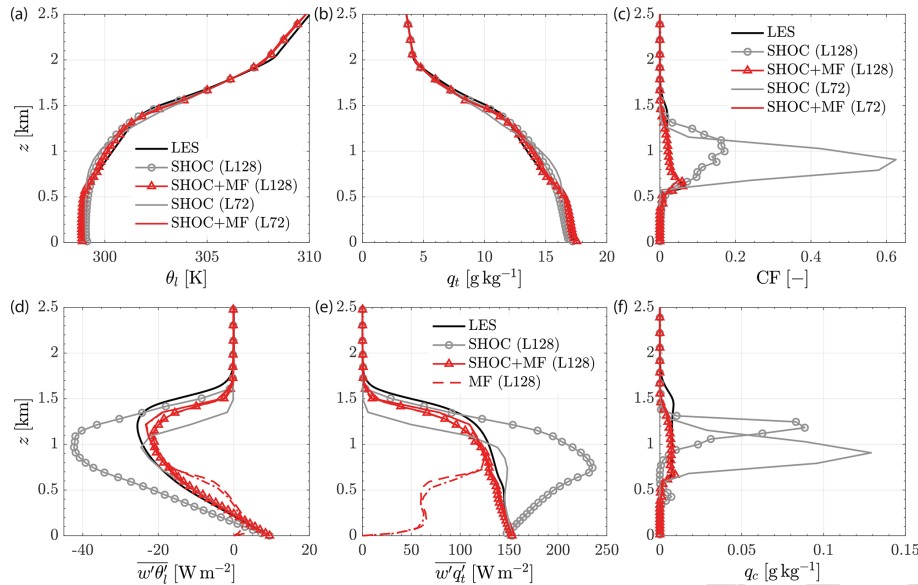

**Figure 8.** Sensitivity of the SCM results to the vertical grid resolution for the BOMEX case. Vertical profiles of **(a)** liquid water potential temperature, **(b)** total water mixing ratio, **(c)** cloud fraction, **(d)** turbulent heat flux, **(e)** turbulent moisture flux, and **(f)** cloud water mixing ratio for LES (solid black line), SHOC (solid grey line), SHOC+MF (solid red line), and MF (dashed red line). The high-resolution vertical grid (L128) profiles are represented by solid lines with markers (circles for SHOC and triangles for SHOC+MF), whereas the coarse-resolution vertical grid (L72) profiles are presented by plain solid lines. The dotted–dashed and dashed red profiles represent the MF contribution using L72 and L128, respectively. The profiles correspond to a time average over $t = 4$–$6$ h.

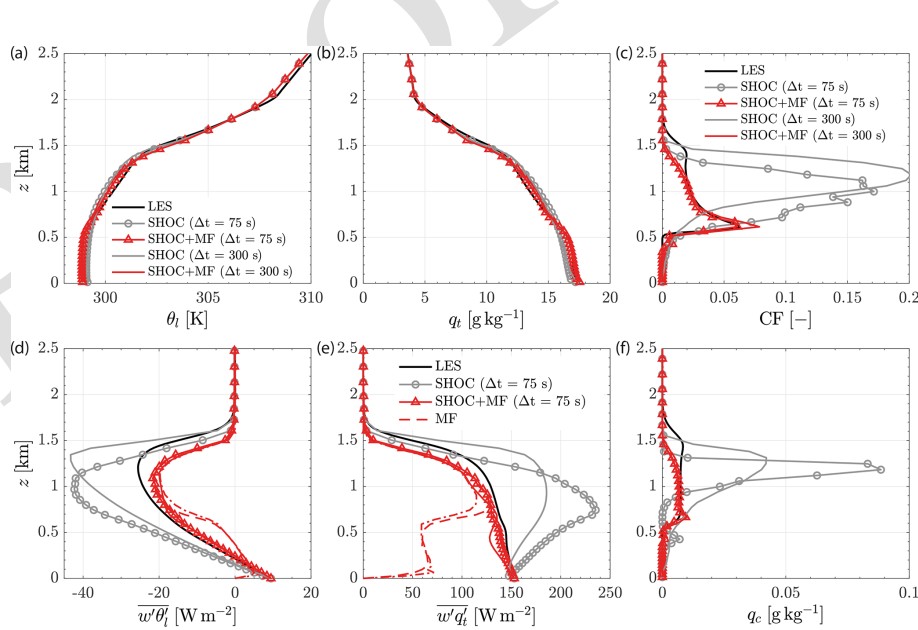

**Figure 9.** Sensitivity of the SCM results to the time step for the BOMEX case. Vertical profiles of **(a)** liquid water potential temperature, **(b)** total water mixing ratio, **(c)** cloud fraction, **(d)** turbulent heat flux, **(e)** turbulent moisture flux, and **(f)** cloud water mixing ratio for LES (solid black line), SHOC (solid grey line), SHOC+MF (solid red line), and MF (dashed red line). The results obtained using $\Delta t = 75$ s are represented by solid lines with markers (circles for SHOC and triangles for SHOC+MF), whereas the results obtained using $\Delta t = 300$ s are represented by plain solid lines. All simulations used the L128 vertical grid. The profiles correspond to a time average over $t = 4$–$6$ h.

cloud fraction values. The cloud fraction values of SHOC surpass those from LES because of the excessive moisture content in the cloud layer (Fig. 4b), and its onset happens about 1 h earlier than in the LES. Conversely, SHOC+MF captures the cloud layer evolution reasonably well due to a better representation of the heat and moisture turbulent transports (Figs. 5 and 6c–d).

### 4.3 Sensitivity to vertical grid resolution and time step

The sensitivity of SHOC and SHOC+MF results to the vertical grid resolution and time step is explored here using the BOMEX case. Results are similar for the ARM case and are thus omitted. We compare the results of BOMEX discussed in Sect. 4.1 using the default vertical grid (72 vertical levels; L72) and a 128-layer vertical grid (L128). This vertical resolution increase translates to about twice as many grid cells within the CBL (including the cloud layer). All other aspects of the model configuration were held unchanged. Lastly, we use the L128 grid to explore the sensitivity of the model to the time step comparing the default 30 min time step to a 75 s time step. The vertical grid and the time step used in SCREAM's global simulations will likely be close to L128 and $\Delta t = 75$ s (Caldwell et al., 2021), thus the importance of exploring the sensitivity of SHOC+MF to both configurations.

Figure 8 shows the sensitivity of the temporally averaged vertical profiles of the thermodynamic and cloud macrophysics variables, as well as turbulent fluxes with respect to the vertical grids. The results of SHOC+MF demonstrate low sensitivity to the grid resolution but still a slight improvement when using L128; e.g., unsurprisingly, the sharp increase in cloud fraction near the cloud base is better resolved with L128 (Fig. 8c although not clearly visible). On the other hand, the results of SHOC show a strong sensitivity to the grid resolution – specifically, both heat and moisture turbulent fluxes roughly double in magnitude in the cloud layer (Fig. 8d–e). On a positive note, this increase in turbulent transport warms up the cloud layer relative to the results using L72, which improves the cloud fraction.

The sensitivity with respect to the time step is shown in Fig. 9. The results of SHOC+MF also show low sensitivity to the time step, while SHOC seems to be slightly sensitive to it. Overall, the results of SHOC+MF do not depend on the vertical grid resolution or on the time step. Thus, further tuning does not seem to be necessary for shallow convection when using a different vertical grid or time step.

### 5 Conclusions

This study documents the implementation of the stochastic-multiplume mass-flux (MF) parameterization (Suselj et al., 2013, 2019a, b) in the Simple Cloud-Resolving E3SM Atmosphere Model (SCREAM) by coupling it to the simplified-higher-order-closure (SHOC) turbulence and cloud macrophysics scheme. The MF contribution to the total turbulent transport is added to SHOC's numerical solver for the moist conserved thermodynamic variables.

SHOC is a unified assumed-PDF-based scheme that represents both boundary layer turbulence and cloud macrophysics, and while it satisfactorily represents dry convection and stratocumulus layers, it struggles to adequately represent shallow cumulus convection (Firl and Randall, 2015; Fitch, 2019). Following a recent study that showed promising results in solving this issue by combining an MF parameterization with the assumed-PDF scheme CLUBB (Witte et al., 2022), we coupled MF to SHOC to improve the representation of shallow cumulus convection in SCREAM.

Our new scheme (SHOC+MF) was evaluated in a single-column-model (SCM) simulation framework against LES reference data for two shallow cumulus convection cases: BOMEX – quasi-steady-state warm maritime shallow convection – and ARM – diurnal cycle of warm shallow convection over land. We also compared the SHOC+MF results with standard SHOC. In general, SHOC+MF represents the mean and flux profiles of moist conserved thermodynamic variables well (liquid water potential temperature, $\theta_l$, and total water mixing ratio, $q_t$), as well as the cloud macrophysics properties (cloud fraction and cloud water mixing ratio, $q_c$) for shallow cumulus boundary layers. This represents an improvement versus SHOC alone, since for BOMEX, SHOC does not reproduce a shallow cumulus layer but rather simulates a structure similar to a stratocumulus boundary layer, and for ARM, SHOC mixes excessively up to 3 km, producing a cloud layer too deep, and also overestimates the cloud macrophysical properties.

We performed a sensitivity analysis to the vertical grid resolution, as well as dynamic and physics time steps for SHOC and SHOC+MF. While SHOC seems to be sensitive to both grid resolution and time step, SHOC+MF showed weak sensitivity to both. Thus, SHOC+MF appears to be robust to changes in the vertical resolution and time step, suggesting there is no need for additional parameter optimization.

In summary, the results of SHOC+MF in SCREAM demonstrate good performance by improving the representation of shallow cumulus convection. Furthermore, the SHOC+MF configuration introduced here is robust enough to properly represent two different shallow cumulus convection cases (i.e., quasi-stationary and non-stationary) regardless of the vertical grid resolution and time step used. Based on these encouraging results, we are currently expanding the evaluation of SHOC+MF to both stratocumulus and deep-convection regimes, as well as to global simulations.

*Code and data availability.* In this study, we used the E3SM model (https://doi.org/10.11578/E3SM/dc.20210927.1, E3SM Project, DOE, 2021), specifically the E3SM SCREAM version dyamond2-try1 released in October 2020 (https:

//github.com/E3SM-Project/scream/releases/tag/dyamond2-try1, last access: 20 March 2023). The modified code shoc.F90 and shoc_intr.F90, the mass_flux.F90, and the scripts and respective IOP files used to generate the present SCM simulations are archived at https://doi.org/10.5281/zenodo.7011628 (Chinita, 2022a). The SCM and LES output data are archived at https://doi.org/10.5281/zenodo.7011652 (Chinita, 2022b).

*Author contributions.* MJC: conceptualization; methodology; formal analysis; visualization. MW: conceptualization; methodology. MJK: conceptualization; methodology. JT: conceptualization; methodology; supervision. KS: conceptualization; methodology. GM: provided the large-eddy-simulation CE9 output data. PB: provided crucial SCREAM code assistance. All authors contributed to interpreting the results, as well as writing and reviewing the manuscript.

*Competing interests.* The contact author has declared that none of the authors has any competing interests.

*Acknowledgements.* Part of this research was carried out at the Jet Propulsion Laboratory, California Institute of Technology, under a contract with the National Aeronautics and Space Administration (80NM0018D0004). We gratefully acknowledge the support of the U.S. Department of Energy, Office of Biological and Environmental Research, Earth System Modeling (DE-SC0019242). This research used resources from the National Energy Research Scientific Computing Center (NERSC), a U.S. Department of Energy Office of Science User Facility located at Lawrence Berkeley National Laboratory, operated under contract no. DE-AC02-05CH11231.

*Financial support.* This research has been supported by the Office of Science (grant no. DE-SC0019242).

*Review statement.* This paper was edited by Simon Unterstrasser and reviewed by three anonymous referees.

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

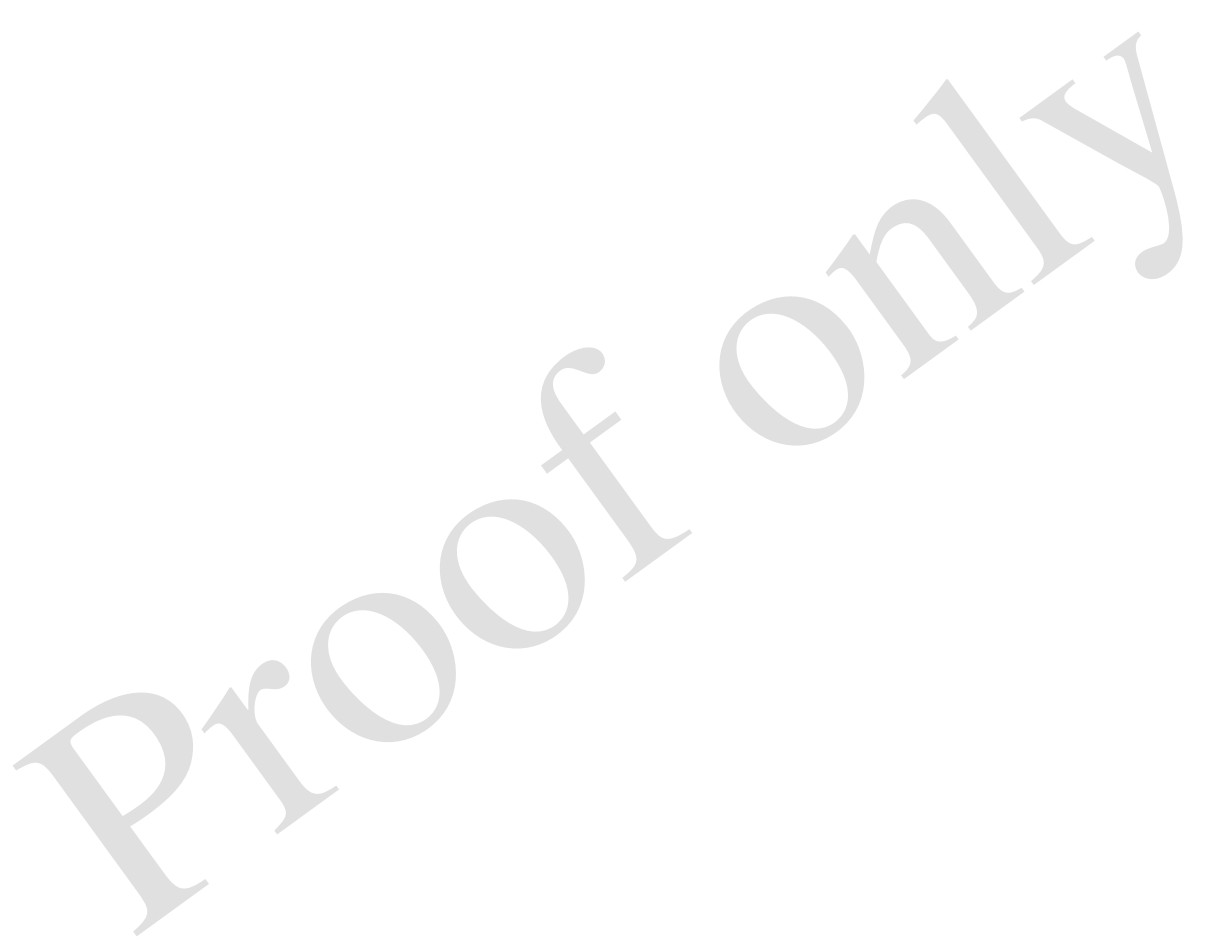

**Remarks from the language copy-editor**

CE1    Your general comment referred the LES instances, but it isn't clear whether you still want to adjust the hyphenation as you said the hyphenation explanation makes sense. You also did not respond to the other instances. I haven't removed the unaddressed comments. Please let me know if any adjustments are necessary or if all instances are okay as currently formatted.

CE2    I've made the change requested in this instance as both forms are possible, but please note that different hyphenation can also convey a different meaning. If you are referring to a model type version that is single column, then this form (changed as requested) is correct. If you are referring to a version of single-column model type, then this should be "single-column-model-type version", as previously formatted.

CE3    As currently hyphenated this refers to the data of the large-eddy simulation. As LES is a compound modifier that modifies data in this case, the entire phrase is hyphenated (e.g., the large-eddy-simulation data vs. the data of the large-eddy simulation). A compound modifier (phrasal adjective) is a phrase that functions as a unit to modify a noun (Chicago p. 227). For a good general overview, see The Chicago Manual of Style, pages 375–382.

CE4    See previous comment for LES on compound modifiers.

CE5    This case is similar to the one above, with SCM modifying mode, so the entire term is hyphenated.

CE6    The hyphenated term above refers to simulations of the single-column model; if this is the intended meaning, then the hyphenation is correct. If you mean model simulations that are single column, then it can be changed to single-column model simulations. Please advise.

CE7    This is another instance of a compound modifier (in this case meaning the forcing files of the intensive observation period; see comment on compound modifiers).

CE8    The instance above is another compound modifier instance referring to the model of the large-eddy simulation (see compound modifier comment).

CE9    This instance is also a compound modifier referring to the output data of the large-eddy simulation.

**Remarks from the typesetter**

TS1    Please confirm subscripts "t", "l" and "u" throughout.

TS2    Please give an explanation of why this needs to be changed. We have to ask the handling editor for approval. Thanks.

TS3    Please give an explanation of why this needs to be changed. We have to ask the handling editor for approval. Thanks.

TS4    Please give an explanation of why this needs to be changed. We have to ask the handling editor for approval. Thanks.

TS5    Please confirm the equation.

TS6    "the" is marked in the current proofreading. Should something be corrected here?

TS7    Please confirm reference list entry.