# Peer review of "Improving the representation of shallow cumulus convection with the Simplified Higher-Order Closure Mass-Flux (SHOC+MF v1.0) approach"

_Geoscientific Model Development, 2022_

## Referee Comment (RC3)

**Initial Review of "Improving the representation of shallow cumulus convection with the Simplified Higher-Order Closure Mass-Flux (SHOC+MF v1.0) approach" by Maria J. Chinita, Mikael Witte, Marcin J. Kurowski, Jaoa Teixeira, Kay Suselj, Georgios Matheou, and Peter Bogenschutz**

November 1, 2022

**1 General Comments**

This manuscript presents a nice discussion of the EDMF and PDF-based HOC PBL and shallow convection literature, describes a new combination of physics schemes to improve on SHOC, and presents relatively preliminary results of using this combination in a SCM to simulate two shallow cumulus cases. It contains excellent writing and grammar and the authors are able to convey their main points in a concise manner, appropriate for the chosen journal. My recommendation would be to accept the paper after minor revisions despite what I perceive as a few major weaknesses of the manuscript, described below. The reason for my recommendation is based on the demonstrated efficacy of the described approach and a recognition that it has *potential* to make a substantial improvement to future GCM/NWP models if further developed.

**2 Specific Comments**

1. I'm concerned that this approach is conceptually "double-counting" effects due to the largest, coherent turbulent eddies. My understanding of the SHOC scheme is that one of the Gaussian components of the underlying PDF is already supposed to account for the type of subgrid-scale PBL-spanning updrafts that the MF scheme is also trying to represent. If I remember correctly, although the complexity of SHOC is boiled down to a K-theory implementation (as evidenced by the first term on the RHS of Eq. 3, 4, 9, 10), several of its underlying assumptions, notably its length scale and the "updraft" portion of the trivariate binormal PDF, are formulated to try to represent the same physical phenomenon that the MF scheme is. I.e., in regions of the column where PBL-scale convective eddies are present, K is substantially increased in SHOC in order to try to represent the effects of those eddies. The need for an additional turbulent transport "boost" from a MF component potentially speaks to the inherent limits of SHOC's approach of maintaining first-order closure (neglecting the TKE component that gives it a 1.5-order closure) at its heart. The manuscript comes close to touching on this point in several places, but never explicitly discusses it. For example, on lines 83-84 where it mentions SHOC represents "local mixing", on lines 197-199 mentioning the down-gradient term from SHOC, and lines 246-248 that describes the modifications to SHOC to achieve satisfactory results. In my reading of this paper, this thought was a through-line that I feel needs to be addressed/discussed. The fact that lines 246-248 are put in the paper lead me to believe that the authors are aware of this issue, but the lack of detail (how was SHOC's length scale reduced, which constant was increased and why, sensitivities to these changes) elicits an impression of "sweeping this under the rug", so to speak. A discussion/explanation doesn't even have to be scientifically/phenomenologically-motivated necessarily. It may suffice just to say that this is a pragmatic approach to patching a "weakness" of the SHOC formulation, etc.

2. The applicability of the paper is limited by the chosen cases. The readers are definitely left "wanting more" in the sense that no indication is given for how the new scheme performs for stratocumulus, deep convection, frontal cloudiness, clear/dry convection, stable PBLs, mixed-phase cloudiness cases, etc. The manuscript as presented is fairly typical in its scope in this regard, and this criticism applies generally to other similar papers, but it is worth pointing out. I'm not saying that the authors need to expand the scope for the particular paper, just that it is much less exciting/convincing without more meteorological regimes (higher "N") to go on.

3. There are components of the scheme that seem arbitrary to the reader. For example, line 144 describing which part of the PDF is used for the MF model with $[1.5\sigma_w - 3\sigma_w]$, or 6.65%. Can this be justified? E.g., why not use the entire part of the PDF with $w > 0$?

4. I'd like a more physical explanation for the formulation of $L_\epsilon$, which I have interpreted as the "mean free path" between entrainment events. It seems like this should be related to the local strength of turbulence, e.g. TKE. I realize that $h_{CBL}$ might be a proxy for TKE, but how it is used seems to be opposite to my intuition. I would think that high TKE (and high $h_{CBL}$) would lead to more frequent entrainment/mixing events, creating a *lower* $L_\epsilon$. This leads me to think that this formulation was "tuned" to get the desired scheme behavior/results rather than using physical reasoning. A more thorough explanation (perhaps it is in the Suselj 2019 paper) for Eq. 8 would be beneficial to the reader.

5. Lines 53-67 discuss EDMF schemes more broadly and the last couple sentences imply that the SHOC + MF approach is on par computationally to existing EDMF implementations. Is this claim accurate? It would be nice to have some numbers to back this up if so.

6. Lines 98-99: Why is C++ code better than Fortran code in this case? Seems strange to make this claim in this paper.

7. Line 115: What are "fluctuations" with respect to? Time, space, both? I'm assuming just space, e.g. fluctuations from the spatial, grid mean values.

8. Line 169: It is perhaps awkward to have two different length scales. How does $L_\epsilon$ related to $L$ used in SHOC?

9. Line 213: I'm confused by this. Doesn't SHOC produce tendencies for $u, v$ like other PBL schemes? Are these not applied too?

10. Line 217: Would it be correct to say that physics then is ONLY using SHOC + MF?

11. Figure 1: Aren't there observations for these case studies? Why are they not plotted alongside LES results?

12. Figure 2: Doesn't SHOC have a binormal PDF? Could you also plots means from the "updraft" component of the PDF? Also, it's explained why there are no data points for SHOC + MF at $z >= 1.5km$, but why wouldn't you plot all lines returning to 0? Shouldn't they all do that at some height?

13. Figure 3: It's pretty hard to see the LES curves. It might be better to plot the bias (difference from LES) for clarity.

14. Figure 6a,b: There are discussions of oscillations like this in the literature. IIRC, Anning Cheng mentions cloud water oscillations related to IPHOC, so it might be worth mentioning that these oscillations are not unique. Also, why do these oscillations not show up in the means in figures 4a,b?

15. Figures 8,9: Same comment about potentially plotting biases for clarity since all lines are on top of each other.

**3    Technical Corrections**

1. Line 77 Instead of "PDFs", it is more appropriate to say Gaussian components or modes. The PDF should mean the entirety (weighted sum of Gaussian components).

2. Line 121: The second approximation is worded strangely in my opinion. You're talking about a small updraft area but the mentioned term doesn't even contain updraft area. I think it would be better to say that the environmental fraction is large such that $w_e \approx \overline{w}$, etc.

---

## Author Comment (AC1)

**Replies to Referee #1 on gmd-2022-162**
The reviewer's comments are shown in bold font.

**The goal of this paper is to test whether the addition of a mass flux scheme, which represent non-local transport by updrafts, to the turbulence scheme called SHOC improves the representation of shallow cumulus convection by the SCREAM climate model.**

**For this purpose, the authors use a single-column model with the SCREAM model physics, and with or without the multiplume mass flux scheme developed by Suselj et. al (2013,2019a,b). By simulating two cloudy boundary layers and comparing them with large-eddy simulations, they show that the addition of this mass flux parameterization improves the representation and evolution of cloud features and turbulent transport.**

**Knowing that uncertainties in the representation of low clouds by climate models remaing important, it is still very valuable to continue working on improving these clouds. The paper presents interesting results, a relevant methodology, and is well written. Therefore, the article is very suitable for publication. I'm nevertheless concerned about the pontential implementation of the SHOC+MF in the parent 3D climate model for several reasons: the adjustement of tunable parameters, the relevance of the EDMF scheme in a model that aims to simulate at high resolution (scale awareness), and the coupling with other parameterization such as radiation, and microphysics. These points are not addressed in this study. More information about would be very valuable for the community.**

We would like to thank the reviewer for the constructive comments that helped to substantially improve the manuscript. Essentially, all of the reviewer's comments were addressed by modifications and additions in the manuscript. As a result of the reviewer's comments, the manuscript was revised.

Detailed replies to the reviewer's comments are listed below.

**Major comments:**
**1. I know that it could be complicated, but I really find it problematic that the authors avoid the question of tuning when testing their new parameterization. Here there are several parameters that need to be ajusted when combining SHOC and MF schemes, e.g. number of plumes N, epsilon_0, 'a' for the entrainment length scale, … The authors justify the tuning parameters' values by the original studies that set them. However, this study highlights a coupling between parameterizations that might suggest that these values may be outdated. Without asking the authors to test all tunable parameters, I would like to know if (1) values they use remain physically consistent in their new SHOCMF framework, and (2) if their conclusions remain similar when some important tunable paremeters are modified. I guess that the second point is feasible given the simple SCM framework the authors use.**

This is indeed an important point. Even though we did not mention it explicitly in the manuscript, we did perform some tuning of SHOC+MF in order to achieve the results discussed in the paper. Our results show a good agreement of SHOC+MF relative to the reference LES for two benchmark cases; this would have not been possible without some tuning. As the reviewer likely knows well,

such tuning is often essential in parameterization implementation and assessment. For the present SHOC+MF configuration, we tested $N$ (number of updrafts), $a_w$ and $b_w$ (constants in the vertical velocity of the $n$th updraft; equation 6 of the paper), $\varepsilon_0$ (fraction of entrained mass-flux during each entrainment event in equation 7), and $a$ (entrainment length scale constant in equation 8). The optimal values $\varepsilon_0 = 0.2$, $a_w = 1$ and $b_w = 1.5$ are equal to the ones used in Suselj et al. 2019, whereas $a = 1.25$ m$^{1/2}$ instead of 2.5 m$^{1/2}$ used in Suselj et al. 2019. Also, here we use $N$ = 40 updrafts. We added a new sentence in section 2.3 discussing the total number of updrafts which reads:

"Here, we use $N$ = 40 updrafts. The number of updrafts was chosen based on a sensitivity analysis of SHOC+MF to its value in which SHOC+MF showed weak sensitivity to $N > 30$ updrafts (not shown). Note that a small number of updrafts can produce noisier results due to the lateral entrainment's stochasticity (Suselj et al. 2019a)."

For the results of SHOC alone, we used the same tunable constants used in the global high-resolution simulation discussed in Caldwell et al 2021 and added a sentence in L249 to clarify: "Lastly, for the simulations of SHOC alone, we used the same tunable constants from the global high-resolution simulation presented in Caldwell et al. (2021)."

**2. The authors show improvements when implementing the MF scheme in SHOC. However, the heat and moisture transports remain biased low (Fig 8-9). Could the authors describe improvements to reduce this bias? Would it be possible to reduce it by a better tuning strategy or switching on radiation scheme?**

Yes, it is likely that we could reduce even further the biases shown by SHOC+MF with additional tuning. However, for the reasons discussed in both major and minor comments #1, it would likely not be necessarily productive at this stage to aim to reduce the already relatively small biases of SHOC+MF shown in Figures 4, 6, 8, and 9.

**Minor comments:**

**1. I understand that removing some parameterization schemes is useful to highlight the novelty of SHOC+MF. Yet I'm surprised that the authors removed the radiation schemes. I would assume that some large-scale cooling and drying forcing are imposed in large-eddy simulations. Removing these schemes would also compromise the ability of the model development to be used in the parent global climate model.**

Because the cases that are investigated are related to shallow convection and as such are related to low values of cloud cover, the LES setup for these cases does not include interactive radiation parameterizations. The low values of cloud cover imply that cloud-radiation interaction plays a secondary role in the overall physics of these cases — as compared to the role of the surface evaporation and of the condensation in the cumulus updrafts. These LES cases do have, however, simple representations of clear-sky radiative cooling that is implicitly part of the large-scale forcing of the LES and the SCM simulations. For the SCM simulations, we also switch off the interactive radiation scheme available in SCREAM because of the reasons explained above and

to allow for a clear comparison between the LES reference data and the SCM results as none of these cases accounts for interactive radiation as mentioned above.

**2. Figures 1+5: I'm confused with the notation. You have two simulations SHOC (A), and SHOC+MF (B). Therefore, you plot results from A and B experiments, but also the relative contribution of SHOC and MF in turbulent fluxes (we can call it B[SHOC] and B[MF]). MF in Figure 1 seems to be B[MF], but this is confusing because MF could also be understood as the difference B-A in Figure 1. I guess that non-linear interactions could make these contributions different (i.e. B-A != B[MF]). Could you clarify this a little bit (differences between experiments vs the MF contribution that is saved from SHOC+MF) ?**

The reviewer's interpretation is correct. For completeness, in figures 1 and 5, we plotted the turbulent fluxes for:
-   SHOC: the turbulent flux diagnosed by the SHOC version; solid gray profiles.
-   SHOC+MF: sum of SHOC and MF following $\overline{w'\phi'} = \underbrace{-K_\phi \frac{\partial \overline{\phi}}{\partial z}}_{SHOC} + \underbrace{\sum_{n=1}^{N} a_n (w_n - \overline{w})(\phi_n - \overline{\phi})}_{MF}$

    (equations 11 in the paper); solid red profiles.
-   MF: the contribution of MF in the SHOC+MF equation above (the term in red): dashed red profiles.

**3. Line: 311: Could you use a single time notation, either "hour XX" or "+YYh" relative to the start of the simulation (as done in Fig 4).**

We edited the labels in Figures 3 and 5. It now reads "HH:MM LST (+X h)", i.e., we changed UTC to local standard time (LST) and followed your suggestion in parentheses. The captions of both figures were updated accordingly.

**4. Figure 8: I don't think the dashed and the dotted-dashed lines are described.**

Thank you so much for catching this. We added its description to the caption of Figure 8.

---

## Author Comment (AC2)

**Replies to Referee #2 on gmd-2022-162**
The reviewer's comments are shown in bold font.

**This manuscript describes the implementation of a stochastic, multi-plume Eddy Diffusivity Mass Flux (EDMF) boundary layer parameterization in the SCREAM GCM, using the Simplified Higher Order Closure (SHOC) to calculate eddy diffusivity and cloud properties. Single column model experiments are used to evaluate the scheme against large eddy simulations for marine shallow cumulus and continental convection cases. Improvements are shown relative to experiments using SHOC alone.**

**The topic is certainly of scientific interest, as boundary layer clouds remain one of the largest sources of uncertainty in future climate projections, and boundary layer parameterization is a topic of active research. The paper is well written and logically organized. However, a few details are missing that are needed for readers to interpret and reproduce results, as noted in comments below. I believe these can be addressed with minor revisions.**

We would like to thank the reviewer for the constructive comments that helped to substantially improve the manuscript. Essentially all of the reviewer's comments were addressed by modifications and additions in the manuscript. As a result of the reviewer's comments, the manuscript was revised. The main changes are:

1. Description of SHOC's turbulence length scale in section 2.4.
2. Description of the coupling of SHOC and MF for the cloud macrophysical properties.
3. Figure 7 was updated following reviewer's comment #1.

Detailed replies to the reviewer's comments are listed below.

**1) There are some discrepancies in cloud top height in the continental convection case between this study, Bogenschutz and Krueger (2013) describing SHOC, and Brown et. al. (2002) documenting the continental case. The two previous papers show LES cloud top heights around 2800 m, while in Fig. 7 here the deepest cloud tops are close to 2400 m. I am wondering if there are differences in case specification that might explain the LES difference, or if it may be due to the use of different LES codes and grid spacings? This has implications for the conclusion that SHOC-MF matches the LES, while SHOC by itself produces too-deep clouds. The original SHOC (Bogenschutz and Krueger, 2013) appeared to match LES in this case fairly well, so the different behavior here warrants some discussion.**

Thank you for bringing this to our attention. In our Figure 7a, the LES cloud top appears to be lower than that in Brown et al. 2002 and Bogenschutz and Krueger (2013; BK13 hereafter) because of the threshold that we applied when producing the figure. In other words, in Figure 7a, we masked CF < 0.01 to eliminate the very small values that may be considered "noise" yielding to what seems to be a considerably lower cloud top. To illustrate this, please see the figure below for the LES cloud condensate (top panel) and cloud fraction (middle and bottom panels):

[Figure]

Figure 1 ARM shallow cumulus case: Time-height plot of LES (a) cloud liquid water mixing ratio, (b) cloud fraction which is defined as $q_l > 1 \times 10^{-5}$ *kg/kg* without any threshold/mask applied, and (c) cloud fraction as in (b) but only the values larger than 0.01 are displayed.

The cloud fraction defined as grid points where $q_l > 1 \times 10^{-5}$ kg/kg is shown in the middle panel, and the cloud fraction field also defined as $q_l > 1 \times 10^{-5}$ kg/kg but masked for CF < 0.01 is shown in the bottom panel—this is the one displayed in Figure 7a.

While our LES cloud-top height is not in disagreement with Figure 5d of Brown et al 2002 where it ranges from 2.4 and 3 km, it is in slight disagreement with Figure 9 of BK13. As the reviewer points out, the LES models and the vertical grid spacing used in both simulations are indeed different—the vertical grid spacing used in BK13 was 40 m, whereas we used 20 m. While we could only be sure that this is the reason for the discrepancy between these LES results by running both LES models with 20 m and 40 m (which is unfeasible), it is fair to conjecture that our cloud-top height is lower due to the higher vertical grid spacing. This is a well-known behavior of LES in which the utilization of higher grid resolutions generally yields slightly shallower boundary layers (e.g., Matheou and Chung, 2014; *Large-Eddy Simulation of Stratified Turbulence. Part II: Application of the Stretched-Vortex Model to the Atmospheric Boundary Layer*).

In conclusion, we replaced figure 7 with a similar figure but using a lower threshold of 0.001 instead of 0.01, and removed "… the cloud top is too deep" in L307.

Lastly, the caption of Figure 7 now reads (new text in blue):
"Figure 7: Time-height plot of (a) LES, (b) SHOC, (c) SHOC+MF cloud fraction for the ARM shallow cumulus case. The LES cloud fraction corresponds to the cloud sampling definition (i.e., $q_l > 1 \times 10^{-5}$ kg kg$^{-1}$). Cloud fraction values < 0.001 are masked and not plotted here."

**2) Additional details are needed regarding the SHOC length scale and how it has been changed since Bogenschutz and Krueger (2013). The length scale formulation can have a major impact on PBL behavior, and should be described in the paper.**

We added SHOC's length scale mathematical description in section 2.4.

**3) It is not entirely clear how convective clouds are treated in this study. Is there any special treatment of cumulus cloud detrained from the mass flux? Or do the updrafts impact clouds only indirectly, through the mean state? This could be noted in Section 2.**

This is indeed an important point. The first thing to note, however, is that for these particular shallow convection case-studies that we are analyzing, the longwave and shortwave radiation parameterizations are off, both in the LES as well as in the SCREAM SCM model. There is a simplified radiation forcing of the LES experiments that is replicated for the SCREAM SCM so as to have the SCM and the LES models forced in the same manner. The key physical reason for why the cloud-radiation interaction is off is related to the fact that with the low values of cloud cover typical of these shallow cumulus case-studies, the expectation is that the cloud-radiation interactions are of secondary importance. Large scale (i.e., in the environment surrounding the updrafts) cloud microphysics is also off, again because the expectation is that large scale microphysics plays a secondary role in these particular case-studies.

In this context, and although important, this topic is of less relevance for these particular shallow cumulus SCM simulations than it would be for other regimes. In any case, and in order to clarify the specific questions from the reviewer: For this version, although there is no explicit detrainment term that is a source of cloud water from the updrafts, the cloud fraction and cloud liquid water that are diagnosed (and shown in the figures) are simple linear combinations of the SHOC (PDF-based) cloud fraction and water with the contributions from the different updrafts.

This is summarized in the paper as follows in section 2.3:

"Condensation in each updraft takes place if the updraft water vapor reaches saturation. The MF contribution to the total cloud fraction corresponds to the sum of the area fraction of the updrafts that condense, and the MF contribution to the total cloud water is defined as the area-average of the cloud water of all moist updrafts."

And in section 2.5:
"The shortwave and longwave radiation, and the large scale cloud microphysics parameterizations are switched off for these experiments—basically because these processes are believed to be of secondary importance for these shallow convection case-studies and as such

are also off for the LES experiments. However, cloud fraction and water are calculated at every model level for diagnostic purposes. In SCREAM, the cloud macrophysical properties cloud fraction and liquid water mixing ratio are estimated using the SHOC pdf. Here, the combination of the grid-mean cloud properties is done as a simple weighted sum of the SHOC and MF contributions:

$$CF = CF_{SHOC} + \sum_{n=1}^{N} a_n (q_{l_n} > 0), \tag{12}$$

$$\overline{q_l} = a_e q_{l_{SHOC}} + \sum_{n=1}^{N} a_n q_{l_n}, \tag{13}$$

where $CF_{SHOC}$ and $q_{l_{SHOC}}$ are diagnosed from SHOC's assumed double Gaussian distribution, and $a_n$ and $q_{l_n}$ are the fractional area and condensate loading of the $n$th updraft. Note that in practice, although the algorithm also imposes $CF \leq 1$, this value is not reached in these simulations because of the low values typical of these shallow convection case-studies. These low cloud values, the overall cloud vertical structure of these shallow convection regimes, and the fact that the radiation parameterization is off, are the key reasons for not using more complex cloud overlap algorithms in these estimates."

**4) The experiments here used N=40 updrafts, while previous papers (e.g., Suselj et al 2013) have typically used a smaller number (N=10). Was N=40 chosen to reduce the effects of stochasticity, and do the results show any sensitivity to the value of N?**

Yes, we choose $N$ = 40 instead of e.g., 10 to reduce the intermittency of the results associated with the stochastic entrainment. Suselj et al. 2019 (*On the Factors Controlling the Development of Shallow Convection in Eddy-Diffusivity/Mass-Flux Models*) showed that using a smaller number of updrafts generally produces noisier results but results converge when a larger number of updrafts is used (please see their section 3a for further details). We tested the impact of $N$ in our SCM experiments and saw reduced sensitivity to the results for $N > 10$ updrafts; please see the figure below with the BOMEX SCM results for $N$ = {10, 40, 50, 100} updrafts.

We updated the text in the paper to reflect this discussion. In section 2.3, around L150, it now reads:

"Here, we use $N$ = 40 updrafts. The number of updrafts was chosen based on a sensitivity analysis of SHOC+MF to its value in which SHOC+MF showed weak sensitivity to $N > 30$ updrafts (not shown). Note that a small number of updrafts can produce noisier results due to the lateral entrainment's stochasticity (Suselj et al. 2019a)."

[Figure]

Figure 2 BOMEX: Sensitivity to number of updrafts *N*: vertical profiles of (a) liquid water potential temperature, (b) total water mixing ratio, (c) cloud fraction, (d) turbulent heat flux, (e) turbulent moisture flux, and (f) cloud water mixing ratio, for LES (black solid line), SHOC+MF (solid lines) and MF (dashed lines) for the BOMEX case.

**5) Are the simple updraft microphysics mentioned on lines 61-64 included in this implementation? If so, do they have a non-negligible impact? This is relevant to the comparison with LES, for which precipitation was disabled.**

No. At this point, we don't have the updrafts coupled to a microphysical scheme in our current SHOC+MF parameterization. Our current implementation reflects Suselj et al. 2019 (*On the Factors Controlling the Development of Shallow Convection in Eddy-Diffusivity/Mass-Flux Models*) as mentioned in L127 but without the downdrafts following the justification in L133-139.

**6) On Line 247, which constant in SHOC was increased to reduce the mixing length, and by how much?**

We increased the tunable constant $l_c$ from 0.5 to 1 in equation 10 (section 2.4) in the SHOC+MF experiments. We updated this sentence to:

"Note that in SHOC+MF, we reduced SHOC's turbulence mixing length scale relative to SHOC alone, to prevent excessive mixing. This was done by increasing the tunable length scale factor $l_c$ from 0.5 to 1 in equation 10. Thus, $l_c = 0.5$ in the SHOC experiments, and $l_c = 1$ in the SHOC+MF experiments."

**L113: Typo: Eqn 1 is missing a "partial" symbol.**

Thank you so much for catching this. It has been fixed.

---

## Author Comment (AC3)

**Replies to Referee #3 on gmd-2022-162**
The reviewer's comments are shown in bold font.

**General comments**

**This manuscript presents a nice discussion of the EDMF and PDF-based HOC PBL and shallow convection literature, describes a new combination of physics schemes to improve on SHOC, and presents relatively preliminary results of using this combination in a SCM to simulate two shallow cumulus cases. It contains excellent writing and grammar and the authors are able to convey their main points in a concise manner, appropriate for the chosen journal. My recommendation would be to accept the paper after minor revisions despite what I perceive as a few major weaknesses of the manuscript, described below. The reason for my recommendation is based on the demonstrated efficacy of the described approach and a recognition that it has potential to make a substantial improvement to future GCM/NWP models if further developed.**

We would like to thank the reviewer for the constructive comments that helped to substantially improve the manuscript. Essentially all of the reviewer's comments were addressed by modifications and additions in the manuscript. As a result of the reviewer's comments, the manuscript was revised.

Detailed replies to the reviewer's comments are listed below.

**Specific Comments**

1. **I'm concerned that this approach is conceptually "double-counting" effects due to the largest, coherent turbulent eddies. My understanding of the SHOC scheme is that one of the Gaussian components of the underlying PDF is already supposed to account for the type of subgrid-scale PBL-spanning updrafts that the MF scheme is also trying to represent. If I remember correctly, although the complexity of SHOC is boiled down to a K-theory implementation (as evidenced by the first term on the RHS of Eq. 3, 4, 9, 10), several of its underlying assumptions, notably its length scale and the "updraft" portion of the trivariate binormal PDF, are formulated to try to represent the same physical phenomenon that the MF scheme is. I.e., in regions of the column where PBL-scale convective eddies are present, K is substantially increased in SHOC in order to try to represent the effects of those eddies. The need for an additional turbulent transport "boost" from a MF component potentially speaks to the inherent limits of SHOC's approach of maintaining first-order closure (neglecting the TKE component that gives it a 1.5-order closure) at its heart. The manuscript comes close to touching on this point in several places, but never explicitly discusses it. For example, on lines 83-84 where it mentions SHOC represents "local mixing", on lines 197-199 mentioning the down-gradient term from SHOC, and lines 246-248 that describes the modifications to SHOC to achieve satisfactory results. In my reading of this paper, this thought was a through-line that I feel needs to be addressed/discussed. The fact that lines 246-248 are put in the paper lead me to believe that the authors are aware of this issue, but the lack of detail (how was SHOC's length scale reduced, which constant was increased and why, sensitivities to these changes) elicits an impression of "sweeping this under the rug", so to speak. A discussion/explanation doesn't even have to be scientifically/phenomenologically-motivated necessarily. It may suffice just to say that this is a pragmatic approach to patching a "weakness" of the SHOC formulation, etc.**

In the introduction, L74-81, and later in L343-347, we briefly address the limitations of double-Gaussian PDF closures in representing the high skewness and kurtosis of the distributions of shallow convection and the need for a larger number of PDFs to properly represent the higher-order moments and cloud statistics. Firl and Randall (2015) illustrate this issue very well in their Figure 6 where four PDFs are needed to properly represent the cloud properties and second-order moments of BOMEX. Regarding this and the "double-counting" issue, Witte et al. 2022 also show in their Figure 3 (see below) the inability of CLUBB (black contours) in representing the extremes of the LES joint distribution (gray contours) and how the MF updrafts successfully capture these extremes in their CLUBB+MF parameterization. It is then reasonable to expect that, if the higher order closures, based on pdfs, such as CLUBB have difficulties in simulating the skewed part of the pdf where most of vertical transport takes place during moist convection, much more simplified pdf approaches such as SHOC will also suffer from (at least) similar problems.

Regarding SHOC, in section 2.4, we added the mathematical description of SHOC's turbulence length scale $L$ (now equation 10) where $l_c$ is a length scale factor that we increase from 0.5 to 1 and that reduces $L$. Please note that, apart from SHOC+MF, this reduction was necessary for the ARM case since SHOC mixes too much in its default configuration (Figures 3–6); however, the reduction of $L$ required to improve SHOC's performance in ARM would degrade even further BOMEX where SHOC is not mixing enough (Figure 1). Overall, we believe the lack of a consistent representation of shallow convection by SHOC is partly due to the general inability of HOC-PDF schemes in representing the shallow convection variability (e.g., Firl and Randall 2015, Fitch 2019, and Witte et al. 2022), although the additional simplifications made in SHOC potentially make the matters worse.

To clarify this, we edited L77-81 and now reads (new text in blue):
"An alternative solution was recently proposed in Witte et al. (2022) where CLUBB is combined with multiple stochastic MF plumes leading to a modified CLUBB+MF parameterization where the plumes represent the extreme tail of the joint distribution which is not represented by CLUBB (see their figure 3). Furthermore, their results showed a large improvement of the higher-order moments for two benchmark shallow cumulus convection cases. Thus, the multiple MF plumes offer a physics-based and cost-effective solution by representing the extreme values of the joint distribution not well captured by the assumed PDF."

[Figure]

FIG. 3. Joint probabilities of perturbation $q'_t$ and $\theta'_l$ relative to level mean $\overline{q_t}$ and $\overline{\theta_l}$ at $t = 6$ h for three levels: (bottom) $z = 250$ m, (middle) $z = 730$ m, and (top) $z = 1000$ m from the BOMEX simulations for LES (gray contours), CLUBB (black contours), and MF plumes (black dots). Red crosses denote mean surface properties relative to the mean environmental properties aloft. Probabilities are expressed as 1 minus a unitless cumulative probability, $C(\theta_l, q_t)$, such that the mean of the joint distributions has a value of 1, decreasing radially outward. The contours intervals are logarithmic and identical for both CLUBB and LES. The outermost contour represents a cumulative probability of 3.5% and the innermost contour represents a cumulative probability of 70%. The CLUBB contours refer to the (left) CLUBB-only and (right) CLUBB+MF simulations, with the black dots denoting the $q'_t$ and $\theta'_l$ values of individual members of the MF plume ensemble.

**2. The applicability of the paper is limited by the chosen cases. The readers are definitely left "wanting more" in the sense that no indication is given for how the new scheme performs for stratocumulus, deep convection, frontal cloudiness, clear/dry convection, stable PBLs, mixed-phase cloudiness cases, etc. The manuscript as presented is fairly typical in its scope in this regard, and this criticism applies generally to other similar papers, but it is worth pointing out. I'm not saying that the authors need to expand the scope for the particular paper, just that it is much less exciting/convincing without more meteorological regimes (higher "N") to go on.**

This is a really important point. As mentioned by the reviewer, this is a fairly typical approach of several similar investigations and papers. To be clear, we are currently extending SHOC+MF to stratocumulus

cases, the stratocumulus to cumulus transition, and the diurnal cycle of convection over land. Our initial focus on shallow convection is essentially because i) these are benchmark case-studies that any successful unified mixing parameterization needs to get right and ii) SHOC does not appear to be able to represent shallow convection cases such as these ones in a realistic manner.

3. **There are components of the scheme that seem arbitrary to the reader. For example, line 144 describing which part of the PDF is used for the MF model with [1.5σw – 3σw], or 6.65%. Can this be justified? E.g., why not use the entire part of the PDF with w > 0?**

We use the updraft model presented in Suselj et al. 2019a; this is mentioned in L127. For clarity, we now reiterate this in L142. We also edited L144 and now reads: "here defined as $1.5\sigma_w < w_n < 3\sigma_w$, where $\sigma_w$ is the vertical velocity standard deviation (note that the interval $[1.5\sigma_w, 3\sigma_w]$ corresponds to a total updraft surface fraction area equal to 6.65% in agreement with the sensitivity analysis to the surface updraft area presented in Suselj et al. (2019a))." Please note that in the EDMF approach, the MF aims to represent the nonlocal turbulent transport by the strongest updrafts. Accordingly, the updrafts correspond to the tail of the joint distribution as shown for instance in figure 3 of Suselj et al. 2013.

Also, we added information about the number of updrafts ($N = 40$ updrafts) around L149 and now reads:
"Here, we use $N$ = 40 updrafts. The number of updrafts was chosen based on a sensitivity analysis of SHOC+MF to its value in which SHOC+MF showed weak sensitivity to $N > 30$ updrafts (not shown). Note that a small number of updrafts can produce noisier results due to the lateral entrainment's stochasticity (Suselj et al. 2019a)."

4. **I'd like a more physical explanation for the formulation of Lϵ, which I have interpreted as the "mean free path" between entrainment events. It seems like this should be related to the local strength of turbulence, e.g. TKE. I realize that hCBL might be a proxy for TKE, but how it is used seems to be opposite to my intuition. I would think that high TKE (and high hCBL) would lead to more frequent entrainment/mixing events, creating a lower Lϵ. This leads me to think that this formulation was "tuned" to get the desired scheme behavior/results rather than using physical reasoning. A more thorough explanation (perhaps it is in the Suselj 2019 paper) for Eq. 8 would be beneficial to the reader.**

Th reviewer is partly correct in that there is, in principle and from a physics perspective, a relation between entrainment and turbulence. However, that precise relation is not really well known (and lateral entrainment parameterizations often reflect this state of knowledge) and in practice the contradiction that the reviewer highlights does not really a play a particularly important role in the context of these specific simulations. While we utilize a stochastic entrainment approach, the foundations in which the parameterization is built are relatively simple and common practice. Equation 8 captures the fact that deeper convective clouds tend to be wider which protects them from the environment leading to small entrainment rates (e.g., Boing et al. 2012, Takahashi et al. 2021). In other words, if the updrafts entrain too often, here through a lower $L_\varepsilon$, they will get diluted and cease too quickly. We edited L170 to clarify this and now reads:
"In agreement with previous studies (e.g., Böing et al. 2012; Takahashi et al. 2021), the entrainment length scale $L_\varepsilon$ is larger for deeper clouds (i.e., higher $h_{CBL}$) as these tend to be wider and thus better

protected from the environment leading to smaller entrainment rates. Note that diagnosing $L_\varepsilon$ as the square-root of $h_{CBL}$ allows for continuous adjustment of $\varepsilon_n$ as a function of the CBL state, i.e., the entrainment rate is reduced for deeper CBLs allowing the updrafts to reach higher vertical levels and vice-versa for shallower CBLs, which is particularly important to represent the strong diurnal cycle over land while remaining insensitive to small oscillations of $h_{CBL}$."

5. **Lines 53-67 discuss EDMF schemes more broadly and the last couple sentences imply that the SHOC+MF approach is on par computationally to existing EDMF implementations. Is this claim accurate? It would be nice to have some numbers to back this up if so.**

This number is largely model-dependent and unfortunately, we don't have an EDMF parameterization, apart from SHOC+MF, available in SCREAM which precludes us from providing meaningful numbers regarding the computational efficiency of EDMF and SHOC+MF. Nonetheless, EDMF and SHOC+MF are significantly more efficient than for instance CLUBB where 9 prognostic equations of higher-order moments must be solved.

6. **Lines 98-99: Why is C++ code better than Fortran code in this case? Seems strange to make this claim in this paper.**

In this paragraph (L94-101), we provide a general description of SCREAM. SCREAM is a global convection-permitting model aiming to run global 3D simulations at a target horizontal resolution of 3 km. The code is being rewritten in C++ to enable faster simulations on GPUs and future architectures than what would be possible using Fortran. The SCREAM model is a fairly recent DOE effort and we expect most readers to be unfamiliar with it, thus a generic introduction seems appropriate.

7. **Line 115: What are "fluctuations" with respect to? Time, space, both? I'm assuming just space, e.g., fluctuations from the spatial, grid mean values.**

In our equation 1, the fluctuations are relative to spatial averages. We added the word "horizontally" to clarify this ("… prognostic horizontally averaged thermodynamic variable…"

8. **Line 169: It is perhaps awkward to have two different length scales. How does Lϵ related to L used in SHOC?**

The reviewer is broadly correct and this is a critical point. Indeed, it can be argued that these two length scales could/should be related. In practice, however, these length scales play different roles and are somewhat independent in SHOC+MF; and EDMF for this matter since most if not all EDMF-type parameterizations contain a length scale equivalent to SHOC's turbulence length scale (as examples see Suselj et al. 2013, 2019; Han and Bretherton 2019; Lopez-Gomez et al. 2020). Developing more realistic links between these length scales so as to better represent in a unified way the key mixing length scales of turbulence and convection, is one of our critical research tasks in moving forward.

9. **Line 213: I'm confused by this. Doesn't SHOC produce tendencies for u, v like other PBL schemes? Are these not applied too?**

That is correct, i.e., SHOC contains 6 prognostic variables ($\theta_l, q_t, TKE, \boldsymbol{u}, \boldsymbol{v}$, and tracer) in the 3D model. However, the single-column model (SCM) used here sets the winds to the values provided in the forcing

file each time the forecast routine is called and unfortunately the forcing file of ARM only contains the initial profiles of *u* and *v*, i.e., it doesn't contain the time-varying vertical profiles of *u* and *v* even though the ARM case represents a diurnal cycle of shallow convection over land. To avoid the wind profiles being "reset" to the initial constant profiles every host model time step, we replaced the wind profiles in the forcing file with the ones from our LES reference run. To be clear, this issue is specific to the ARM case and it is because and citing Brown et al. 2002: "At the time when the case was set up, detailed estimates of the time and height variation of the geostrophic wind, and of any large-scale advective tendencies of the wind components, were not available."

This paragraph was rewritten and reads:
"It is worth noting that we modified the ARM case forcing file to run the model with a 30-minute time step (i.e., the ARM forcing file available in the E3SM SCM library contains values at every 20 minutes). Also, the SCM reads the wind information from the forcing file at every host model time step, however for the ARM case, the large-scale advective tendencies of the winds were not available when the case was setup (Brown et al. 2002), and consequently, the time-varying u profiles in the forcing file were set equal to the initial profiles which are constant with height. Resetting the u profile to the initial vertically constant profile at every host model time step interferes with the development of the TKE field through the shear production term. To circumvent this issue, we replaced the u profiles in the forcing file with the u profiles from our LES reference data; the meridional wind component v is zero in the ARM case. Note that this is specific to the SCM used here and to the ARM case as the large-scale advective tendencies of the winds were not available when the ARM case was setup (Brown et al. 2002)."

**10. Line 217: Would it be correct to say that physics then is ONLY using SHOC + MF?**

Yes, that is correct. But please note that (i) both BOMEX and ARM correspond to non-precipitating warm shallow convection, so cloud microphysics does not play a key role; (ii) the cloud cover is low so cloud-radiation interactions are not expected to play an important role and (iii) and that the clear-sky radiative forcing is included in the 'SCM forcing file'.

**11. Figure 1: Aren't there observations for these case studies? Why are they not plotted alongside LES results?**

Both shallow convection cases are idealized case studies based on observations in which the large-scale forcings represent a spatial average of the BOMEX square ($500 \times 500$ km$^2$) and the ARM Southern Great Plains site (140 000 km$^2$), respectively, instead of local forcings. Accordingly, there are no direct observations to compare the modeling results. Further details on this are available in Siebesma and Cuijpers (1995) ("Evaluation of Parametric Assumptions for Shallow Cumulus Convection") and Brown et al (2002) ("Large-eddy simulation of the diurnal cycle of shallow cumulus convection over land"). Nevertheless, LES provides an explicit solution of the fundamental fluid flow equations given the prescribed initial and boundary conditions. As the reviewer likely knows well, SCM comparisons against LES output of benchmark cases are common practice in numerous studies. For example, the Siebesma et al. (2003) paper entitled "A large eddy simulation intercomparison study of shallow cumulus convection" has over 700 citations.

**12. Figure 2: Doesn't SHOC have a binormal PDF? Could you also plots means from the "updraft" component of the PDF? Also, it's explained why there are no data points for SHOC + MF at z >=1.5km, but why wouldn't you plot all lines returning to 0? Shouldn't they all do that at some height?**

Yes, SHOC has a binormal PDF. Please, see below the properties of SHOC's "updraft" Gaussian (dashed gray line with asterisks). The updraft contribution of SHOC's gaussian is very small and ceases around 800 m as expected based on Figure 1 where the MF contribution takes over in the cloud layer and becomes the only contribution to the total fluxes above 800 m (see the MF contribution in Figure 1d and e; dashed red line).

Regarding SHOC+MF, we thank the reviewer for pointing this out. The SHOC+MF data points stop at the cloud top where the updrafts cease. Please note that the hourly average vertical profiles in panels c and d correspond to $\theta_{l_u} - \theta_l$ and $q_{t_u} - q_t$, respectively; we applied a mask to the data so that grid points where there are no moist updrafts are not included in the hourly average and the difference calculation relative to the mean fields. Figure 2 in the paper has been updated.

To keep Figure 2 as simple and easy to interpret as possible, we did not add SHOC's gaussian contribution, instead we added the following explanation in L276:
"Note that the "updraft" (second Gaussian) moist properties of the SHOC's PDF are not shown because they are quite small (e.g., maximum $w_u \approx 0.3$ m/s) and vanish around 800 m in agreement with Figure 1d-e where the MF contribution makes up the total turbulent fluxes."

[Figure]

Figure: Vertical profiles of moist updraft properties for the BOMEX case. (a) Updraft area, (b) updraft vertical velocity, and excess relative to the grid-mean values of (c) liquid water potential temperature, $\theta_{lu} - \bar{\theta}_l$, and (d) total water mixing ratio, $q_{tu} - \bar{q}_t$. The solid and dashed black lines correspond to the LES cloud sampling and the cloud core sampling, the dashed gray lines to the SHOC's contribution of SHOC+MF (i.e., the properties from the "updraft" Gaussian of SHOC), and the solid red lines to the MF contribution of SHOC+MF. The profiles correspond to a time average over $t$ = 4–6 h.

**13. Figure 3: It's pretty hard to see the LES curves. It might be better to plot the bias (difference from LES) for clarity.**

These biases (SCM—LES) are shown in figure 4, and same for figures 5 and 6 but as time-height curtain plots since ARM is a non-stationary case.

**14. Figure 6a,b: There are discussions of oscillations like this in the literature. IIRC, Anning Cheng mentions cloud water oscillations related to IPHOC, so it might be worth mentioning that these oscillations are not unique. Also, why do these oscillations not show up in the means in figures 4a,b?**

Thanks for bringing this paper to our attention. At a first glance, these oscillations seem to be unrelated to the ones described in Cheng 2004 where the "liquid water oscillations are caused by the interaction of the LWB and the mean gradient of $\bar{s}$ in the third-order equations" (LWB: liquid water buoyancy terms). Here, these oscillations appear to be related to the eddy turnover timescale $\tau$ used in SHOC's turbulence length scale (please see now equation 10 in section 2.4) when using its dynamic definition instead of a constant value (in the current development codebase of SCREAM, $\tau = 400$ s).

Lastly, these oscillations show up in figures 4a and b but they are only visible if we increase the colormap limits; however, we would prefer to have the same colormap limits on both the top and bottom panels for comparison purposes.

**15. Figures 8,9: Same comment about potentially plotting biases for clarity since all lines are on top of each other.**

We appreciate the reviewer's comment but we believe our message comes across as effectively through our current figures. Figures 8 and 9 are for BOMEX which contrarily to ARM (figures 3–4 and 5–6 where 4 and 6 show the biases over time of the profiles shown in figures 3 and 5) is a steady-state case. Ultimately, the lines being on top of each other is somewhat the goal here as it means that the differences between the reference LES and the SCM are negligible.

**Technical Corrections**

**1. Line 77 Instead of "PDFs", it is more appropriate to say Gaussian components or modes. The PDF should mean the entirety (weighted sum of Gaussian components).**

Thank you for pointing it out. We fixed this sentence and now reads:
"… higher-order moments and cloud statistics appear to only be properly represented when a larger number of Gaussians is used in the joint PDF"

**2. Line 121: The second approximation is worded strangely in my opinion. You're talking about a small updraft area but the mentioned term doesn't even contain updraft area. I think it would be better to say that the environmental fraction is large such that we ≈ w, etc.**

Thanks. We rephased this sentence and now reads:
"… the second term is neglected because the environmental and grid-mean values are approximately equal (i.e., $w_e \approx \bar{w}$) following the assumption of small updraft area (i.e., $a_u \ll 1$)".